# Wafer-scale manufacturing of ultra-broadband, high-power erbium-doped integrated lasers

Xinru Ji [1,2], Xuan Yang[1,2], Yang Liu [1,2], Zheru Qiu [1,2], Grigory Lihachev[1,2,4], Simone Bianconi [1,2], Jiale Sun[1,2], Andrey Voloshin [1,2], Taegon Kim[3], Joseph C. Olson[3] & Tobias J. Kippenberg [1,2] ✉

Erbium (Er) is an attractive gain medium for amplifiers and lasers due to its long excited-state lifetime, low noise and nonlinearity, and temperature stability. Recently developed ultra-low-loss $Si_3N_4$ photonic integrated circuits combined with Er ion implantation have enabled high-performance on-chip Er lasers, but manufacturing scalability has been limited by the high 2 MeV implantation required for tightly confined 700-nm-thick waveguides. Here we demonstrate the first fully wafer-scale, foundry-compatible Er-doped $Si_3N_4$ tunable lasers by using 200-nm-thick waveguides, reducing implantation energy to below 500 keV and enabling usage of 300-mm industrial implanters. The low-confinement design also improves laser performance and output power. We achieve 91 nm tuning across the C- and L-bands, 47.6 mW fiber-coupled output power, and a 78.5 Hz intrinsic linewidth. Devices operate up to 125°C and show less than 15 MHz drift over 6 hours, enabling scalable high-performance Er-doped lasers for integrated photonics.

Erbium-doped fiber lasers (EDFLs)[1–3] are regarded as the benchmark for lowest laser phase noise, and play a key role in a wide range of applications such as distributed fiber sensing[4,5], gyroscopes[6], free-space communications[6], and optical frequency metrology[7]. Their advantages are attributed to the unique properties of Er ions, including a long excited-state lifetime, slow gain dynamics, temperature stability, and low noise figure[8]. However, their bulky size, high cost from manual assembly and system complexity have relegated Er-doped fibers predominantly to the laboratory environments and limited industrial use where the footprint is less critical. The realization of integrated lasers based on Er-doped photonic waveguides − using Er ions as the same gain basis as in fiber lasers − offers the potential for device miniaturization, fiber laser coherence, and temperature insensitivity in a monolithic architecture. Nevertheless, hybrid Er-based integrated lasers have historically underperformed compared to commercial fiber lasers in terms of intrinsic linewidth and output power[9–17], until recently: direct Er ion implantation into

ultra-low loss $Si_3N_4$ waveguides has enabled the realization of Er-doped waveguide amplifiers[18] and has further led to the development of Er-doped waveguide lasers (EDWLs)[19], which match the coherence of commercial Er-doped fiber lasers while surpassing them in tunability. These achievements were realized in 700 nm-thick, tightly confined $Si_3N_4$ waveguides. However, such thick $Si_3N_4$ waveguides require high implantation energies (up to 2 MeV) to achieve sufficient ion penetration for optimal overlap with the optical field. This significantly limits system scalability, as it demands specialized implantation equipment that has small area coverage (typically $<2 \times 2\,cm^2$), low beam current, long processing times, and introduces challenges such as heating and waveguide deformation from high-energy ion beams[18]. Prior wafer-scale rare-earth lasers deposited $Er:Al_2O_3$ on Si or $Si_3N_4$ waveguides[20,21], and mode-hybrid $Tm:Al_2O_3/Si_3N_4$ large mode area (LMA) amplifiers and lasers have scaled on-chip power[22,23], but neither is a monolithic, directly implanted $Er:Si_3N_4$ platform.

[1]Institute of Physics, Swiss Federal Institute of Technology Lausanne (EPFL), Lausanne, Switzerland. [2]Institute of Electrical and Micro Engineering, Swiss Federal Institute of Technology Lausanne (EPFL), Lausanne, Switzerland. [3]Varian Semiconductor, Applied Materials, Gloucester, MA, USA. [4]Present address: EDWATEC SA, Lausanne, Switzerland. ✉e-mail: tobias.kippenberg@epfl.ch

Here, we directly implant Er into $Si_3N_4$ waveguides and demonstrate wafer-scale, C+L-band tunable waveguide lasers with fiber-laser-class coherence without external seeding[24]. This is realized through a low-confinement $Si_3N_4$ platform with a 200 nm waveguide thickness, which substantially reduces the required ion beam energy to below 500 keV. This approach ensures compatibility with standard industrial implanters for 8- to 12-inch wafers (Fig. 1H), improving cost-efficiency, throughput, and fabrication scalability of Er-doped $Si_3N_4$ (Er:$Si_3N_4$)

devices using established semiconductor protocols[25,26]. It reduces Er implantation time from tens of hours for a $2 \times 2\,cm^2$ area to tens of minutes for 12-inch wafers (Supplementary Note 1), minimizes waveguide deformations compared to high-energy methods[18], as verified by cross-sectional SEM image in Fig. 1E, thereby bridging research-grade and industrial-scale fabrication. A detailed comparison of Er ion implantations in high- and low-confinement $Si_3N_4$ waveguides is provided in Supplementary Note 1. Furthermore, the low-confinement

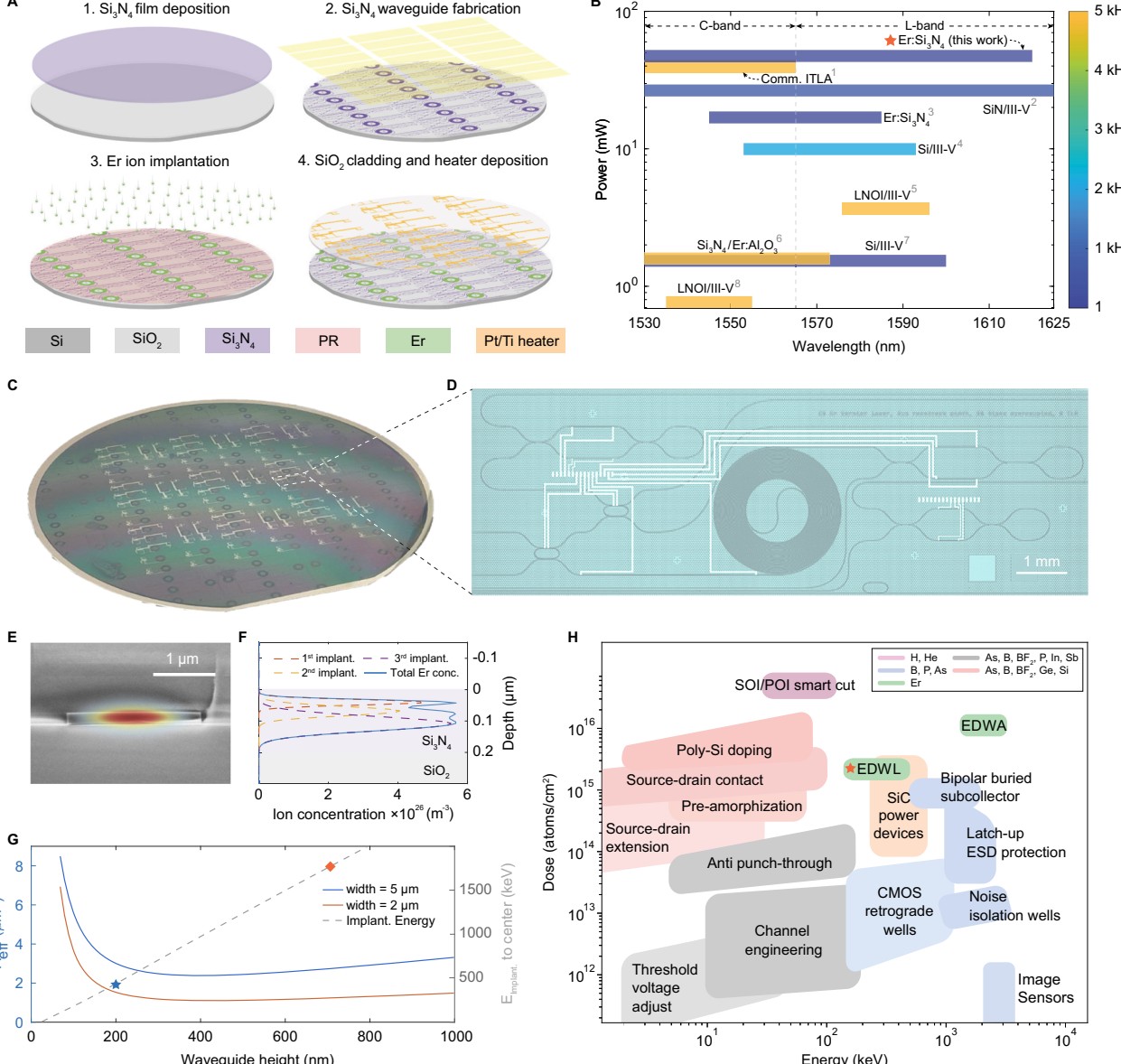

**Fig. 1 | Wafer-scale manufacturing of Er-doped $Si_3N_4$ (Er:$Si_3N_4$) photonic integrated circuits. A** Schematic of key processes for wafer-scale fabrication of Er:$Si_3N_4$ PICs, including 4-inch $Si_3N_4$ photonic wafer fabrication, Er ion implantation, and micro-heater deposition. A 6 μm-thick photoresist mask (AZ 15nXT), patterned via laser writing, enables wafer-scale selective Er ion implantation to $Si_3N_4$ spiral waveguides serving as gain sections. **B** Comparison of key performance metrics for state-of-the-art integrated tunable lasers, including wavelength tuning range, output power, and intrinsic linewidth. References for the compared devices: Comm. ITLA (1): Lumentum ITLA; SiN/III-V (2):[27]; Er:$Si_3N_4$ (3):[19]; Si/III-V (4):[28]; LNOI/III-V (5):[29]; $Si_3N_4$/Er:$Al_2O_3$ (6):[21]; Si/III-V (7):[30]; LNOI/III-V (8):[31]. **C** Optical image of a 4-inch wafer of Er:$Si_3N_4$ integrated tunable lasers following the fabrication processes outlined in **A**. **D** Optical image of a single Er:$Si_3N_4$ integrated tunable laser with a footprint of $0.4 \times 1.0\,cm^2$. **E** Scanning electron microscope image of a 200-

nm-thick Er:$Si_3N_4$ waveguide overlaid with a simulated fundamental transverse electric (TE) optical mode. **F** Simulated erbium ion distribution along the vertical direction of a 200-nm-thick $Si_3N_4$ waveguide, based on three consecutive ion implantation steps with a maximum beam energy of 480 keV, optimized for overlap between Er ions and the optical mode. **G** Simulated effective optical mode areas of the fundamental TE mode and required implantation beam energy to reach the waveguide center, as a function of the $Si_3N_4$ waveguide thickness. For comparison, the implantation energies for 700 nm[19] and 200 nm-thick waveguides (this work) are indicated. **H** Implantation energy and dose of commonly used ion implantation processes in the microelectronics industry[25,26], compared to the energy and dose required for previous implementations of Er-doped waveguide amplifier (EDWA)[18] and Er-doped waveguide laser (EDWL)[19], as well as for the EDWL implementation presented in this work (marked with red star).

design mitigates optical nonlinear effects that degrade laser performance. Collectively, we present hybrid integrated erbium-based lasers with a compact $0.4 \times 1.0$ cm² footprint, a 91 nm wavelength tuning range spanning nearly the entire C- and L-bands, and fiber-coupled output power up to 47.6 mW and an intrinsic linewidth of 78.5 Hz. This outperforms the power and tuning range of commercial Er-doped fiber lasers and III-V based integrated lasers (Fig. 1B,[21,27–31]), paving the way for scalable, cost-effective manufacturing of rare-earth-doped photonic integrated circuits, providing affordable high-coherence light sources for a wide range of applications.

## Results

### Wafer-scale ion implantation and manufacturing of Er-doped photonic integrated circuits

The wafer-scale manufacturing process of the EDWL starts with fabricating ultra-low loss Si₃N₄ photonic integrated circuits (Fig. 1A). The Si₃N₄ waveguides are formed from 200 nm low-pressure chemical vapor deposition (LPCVD) Si₃N₄ films on Si wafers with an 8 μm wet-oxidized SiO₂ layer. Post-deposition annealing at 1200 °C eliminates hydrogen-related defects[32], minimizing absorption in the 1500–1530 nm range[33,34]. Waveguides are patterned with deep ultraviolet (DUV) stepper lithography and dry-etched to achieve smooth and vertical sidewalls. Details on the ultra-low-loss Si₃N₄ PIC fabrication are provided in the "Methods" and Supplementary Note 2.

Prior to the Er ion implantation, a 6 μm-thick photoresist layer (AZ 15nXT) is applied to selectively expose regions for Er implantation while shielding passive areas. Er ion implantation is performed on passive Si₃N₄ waveguides using a commercial VIISta HE ion implanter, capable of processing wafers up to 300 mm in diameter and supporting simultaneous implantation into four 4-inch wafers used in this work. To maximize the overlap between the erbium ions and the fundamental TE optical mode while minimizing parasitic upconversion, three consecutive implantation steps are carried out at energies of 480, 270, and 130 keV, achieving a homogeneous ion distribution. The optimized ion fluences for each step are $3.2 \times 10^{15}$, $1.5 \times 10^{15}$, and $1.1 \times 10^{15}$ cm⁻², respectively, resulting in a total implantation time of 100 min (details in Supplementary Note 1). Figure 1F shows the simulated erbium ion concentration profile using the Monte Carlo program package "Stopping and Range of Ions in Matter" (SRIM[35]), with a maximum projected range over 100 nm, achieving an overlap of $\Gamma \approx 30\%$.

After ion implantation, the wafer is annealed at 1000 °C for 1 h in N₂ to optically activate erbium ions and heal implantation-induced defects. A 3 μm hydrogen-free, low-loss SiO₂ cladding is deposited using SiCl₄ and O₂ precursors at 300 °C, thereby minimizing O–H absorption[36] and precluding SiOₓNᵧ formation[37]. Platinum (Pt) and titanium (Ti) micro-heaters are then fabricated for thermo-optic tuning.

### Hybrid integrated erbium-based Vernier laser

The erbium-based laser (Fig. 2A) comprises a linear optical cavity with an Er-doped gain section between two tunable loop mirrors, and a Vernier-mode filter formed by two microresonators with slightly different free spectral ranges (FSRs). Figure 2B presents the photoluminescence (PL) spectra of Er-implanted Si₃N₄ and SiO₂ thin films, excited by a 520 nm diode laser which populates the $^4I_{13/2}$ state via non-radiative decay from higher energy levels, avoiding in-band stimulated emission. The area-normalized and scaled PL intensities indicate that the profile and bandwidth of Er ions in Si₃N₄ closely resemble those in SiO₂ optical fiber. The relative intensities of the two main PL peaks are attributed to variations in the local crystal field.

The erbium ions in the integrated laser platform can be optically pumped via two approaches: edge coupling with a 1480 nm III-V laser diode (3SP 1943LCV1) or remote pumping using a high-power fiber-coupled module (QPhotonics QFBGLD-1480-500). The edge couplers

have a height of 200 nm and widths tapering from 0.42 μm at the input side and 0.5 μm at the output side. Simulations indicate an insertion loss of 0.97 dB for the input coupler when edge-coupled to the 1480 nm pump diode and 0.56 dB when butt-coupled with UHNA-7 fibers.

An intra-cavity Vernier ring filter comprising two cascaded add-drop racetrack microresonators (Fig. 2A) is implemented to achieve laser mode selection. To ensure single-mode operation and minimize transmission losses to higher-order transverse modes, both resonators feature a waveguide width of 2 μm and a 3 dB bandwidth narrower than the laser cavity longitudinal mode spacing of 300 MHz. The microresonators designed with FSRs of 144 GHz ($\approx$1.15 nm) and 142 GHz ($\approx$1.14 nm) result in a Vernier FSR of 10.224 THz ($\approx$81.9 nm). The broadband transmission measured at the drop-port of two Vernier resonators (Fig. 2C), along with the frequency differences between neighboring resonances (in red), confirms a Vernier filter spacing of $\approx$10 THz, covering the major Er emission band. Figure 2D presents the TE-mode propagation loss of a passive microring resonator with a 50 GHz FSR and 5 μm width, measured by scanning laser spectroscopy after the complete process flow, reflecting the baseline passive platform loss. Figure 2E shows a representative optical resonance with a fitted intrinsic linewidth of $\kappa_0/2\pi = 22.6$ MHz. $\alpha$ (dB/m) is derived from the fitted intrinsic resonance loss $\kappa_0/2\pi$ using $\alpha = 10\log_{10}(e) \cdot n_g\kappa_0/c$, where the group index $n_g = 1.8$ and $c$ is the speed of light. The waveguide propagation loss $\alpha$ varies between 1.5 and 4 dB/m across the C- and L-bands, with wavelength dependence primarily attributed to Rayleigh scattering at the waveguide sidewalls.

Pt/Ti micro-heaters are employed to tune the laser emission wavelength by aligning the peak transmission of the Vernier filter with a cavity longitudinal mode, while the Er:Si₃N₄ gain spiral remains near ambient during tuning. Consequently, temperature-induced Er³⁺ population redistribution[8] is negligible and does not determine the tuning range. The tuning efficiency of the micro-heater is characterized in Fig. 2F, where the optical spectrum map demonstrates wavelength tuning of filtered amplified spontaneous emission (ASE) measured at the drop-port of the Vernier resonator. Single-mode linear fine tuning is observed as the heater power increases from 0 to 400 mW, achieving a continuous tuning efficiency of 620.6 GHz/W. The tunable broadband cavity reflectors use a looped Mach–Zehnder interferometer (MZI) structure. Directional couplers within the MZI regulate power distribution between the two arms, determining the mirror output. Phase tuning is achieved via micro-heaters, which introduce precise phase shifts to flexibly control the broadband transmission and reflection characteristics of the loop mirrors (Fig. 2G). A comprehensive analysis of the loop mirror design and tuning performance of fabricated devices is provided in Supplementary Note 3, demonstrating full transmission and reflection under specific bias conditions.

### Laser wavelength tuning and emission coherence

We investigated the EDWL lasing performance and coherence using the experimental setup illustrated in Fig. 3A. To minimize external disturbances and validate laser performance, we performed photonic packaging in a custom 14-pin butterfly package (Fig. 3B). A Peltier element, a thermistor, and all micro-heaters are connected to butterfly pins using wire bonding. Erbium ions in the packaged device are optically excited from $^4I_{15/2}$ to $^4I_{13/2}$ level by in-band pumping from a 1480 nm laser diode (QPhotonics QFBGLD-1480-500, >1 nm spectral width near 1480 nm) with 400 mW nominal power at 1.5 A driving current. The laser input and output waveguides were edge-coupled and glued using cleaved UHNA-7 optical fibers spliced to SMF-28 fiber pigtails, resulting in insertion losses of 1.62 dB at the input and 1.28 dB at the output at 1550 nm (Supplementary Note 4). The optical spectrum in Fig. 3C shows single-mode lasing at 1561.8 nm with 47.6 mW output power in fiber and an 80 dB side mode

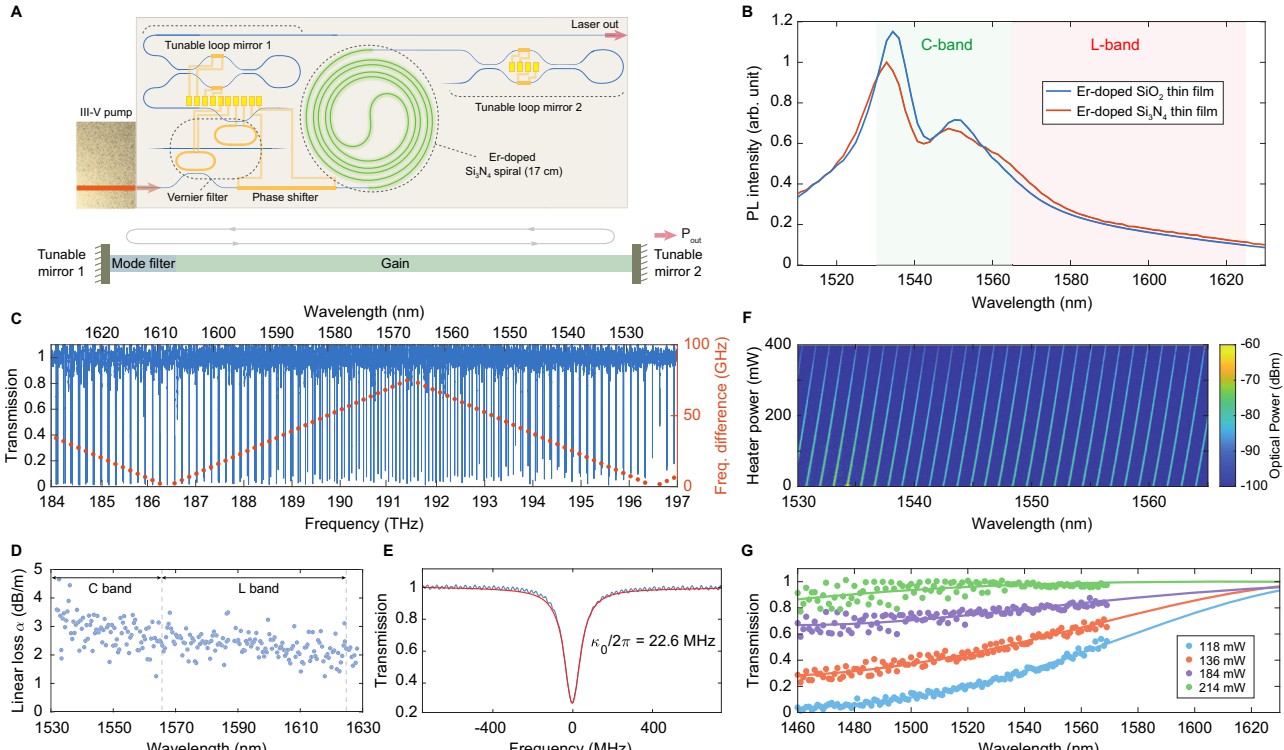

**Fig. 2 | Characterization of volume-manufactured Er:Si₃N₄ integrated laser components. A** Schematic of a hybrid integrated tunable Er laser featuring a microresonator-based Vernier filter for wavelength tuning, broadband loop mirrors for tunable mode reflection, and an Er:Si₃N₄ gain section. **B** Measured photoluminescence (PL) spectra of Er-doped Si₃N₄ and SiO₂ films, pumped by a 520 nm diode laser to excite the $^4I_{13/2}$ state via decay from higher energy levels, free from in-band stimulated emission. Both PL intensities are area-normalized and scaled for clarity. **C** Broadband optical transmission measured at the drop-port of two Vernier microresonators with slightly differing FSRs. The resulting Vernier spacing is approximately 10 THz, corresponding to an 82 nm wavelength range.

**D** Propagation loss of passive Si₃N₄ waveguides, obtained from the intrinsic linewidths of a 50 GHz FSR microring resonator with 5 μm width and 200 nm height. **E** A measured optical resonance (blue) with a fitted intrinsic linewidth $\kappa_0/2\pi$ of 22.6 MHz (red). **F** Thermo-optical tuning characterization of the integrated heater on a Vernier microresonator. The optical spectrum illustrates the wavelength tuning of filtered amplified spontaneous emission (ASE) as the heater power increases from 0 to 400 mW. **G** Measured (dots) and simulated (solid lines) transmission of a broadband tunable loop mirror. The transmission response is thermally tuned by a micro-heater.

suppression ratio (SMSR) with 0.02 nm analyzer resolution bandwidth. We observe no four-wave mixing (FWM) sidebands in current devices, including at the highest output power of 47.6 mW (Fig. 3C). Although Si₃N₄ resonators with high Q can in principle support FWM due to strong intracavity field buildup, the strong normal group-velocity dispersion and LMA of the 200 nm waveguides prevent efficient phase matching and thus suppress nonlinear interactions under our operating conditions. Figure 3D demonstrates the off-chip single-mode laser tuning across a broad wavelength range, from 1530 to 1621.3 nm. In this measurement, the Er:Si₃N₄ waveguides were bidirectionally pumped via a fiber array designed to match the spacing between the input coupler and the mode-selective loop mirror, which transmits 1480 nm pump light while reflecting the 1550 nm lasing wavelength (Supplementary Note 3). Such pumping configuration sufficiently excites Er ions to the $^4I_{13/2}$ state across the entire gain waveguide. Laser wavelength tuning is realized by adjusting the heater power on one microresonator in the Vernier filter, aligning distinct cavity modes with the filter passband. To maximize output power at the target wavelength, the phase shifter inside the laser cavity and loop mirrors were adjusted during the optimization process. The discrete lasing in Fig. 3D corresponds to the FSR of the Vernier microresonator. Although the laser is not strictly mode-hop-free (i.e., it does not continuously tune a single longitudinal mode), simultaneous tuning of both rings enables continuous[38] and deterministic single-mode coverage across the C + L band through Vernier selection, without uncontrolled mode hopping.

This laser tuning exceeds the capabilities demonstrated in previous integrated laser systems[19,21,28–31], achieving performance comparable to that of benchtop EDFLs optimized for EDF length and intra-cavity loss[39]. Nevertheless, a decrease in C-band power relative to the L-band is observed despite stronger Er emission near 1530 nm. This is primarily due to the limited Vernier FSR (≈80 nm; Fig. 2C), which co-aligns C- and L-band resonances (e.g., 1530/1610 nm) and biases lasing toward the L-band where passive loss and absorption are lower, yielding higher net gain $g-\alpha$. The near-full C+L lasing in Fig. 3D is obtained by bidirectional pumping to raise inversion and by biasing the WDM loop mirror to suppress L-band modes; however, this also increases the loss seen by co-aligned C-band modes, limiting the output power. A wavelength-resolved model in Supplementary Notes 5 and 6, including $\alpha(\lambda)$, Vernier transmission, and ion-pair quenching[40], quantifies the measured power trend with wavelength. In the ideal limit (no quenching, wavelength-independent loss, unity Vernier transmission), the EDWL demonstrates near-uniform spectral behavior. Therefore, to enhance C-band output, we target reduced short-wavelength loss, a larger Vernier FSR and transmission, and a lower effective pair fraction (e.g., reduced Er dose with longer gain length). Figure 3E presents the on-chip laser power as a function of on-chip pump power, revealing a lasing threshold of 17.5 mW at 1480 nm pumping and a slope efficiency of ~24% at 1591.9 nm. Supplementary Note 6 presents simulations and measurements of slope efficiency as a function of wavelength, showing higher efficiency across 1560–1600 nm rather than at the Er³⁺ emission peak, due to reduced

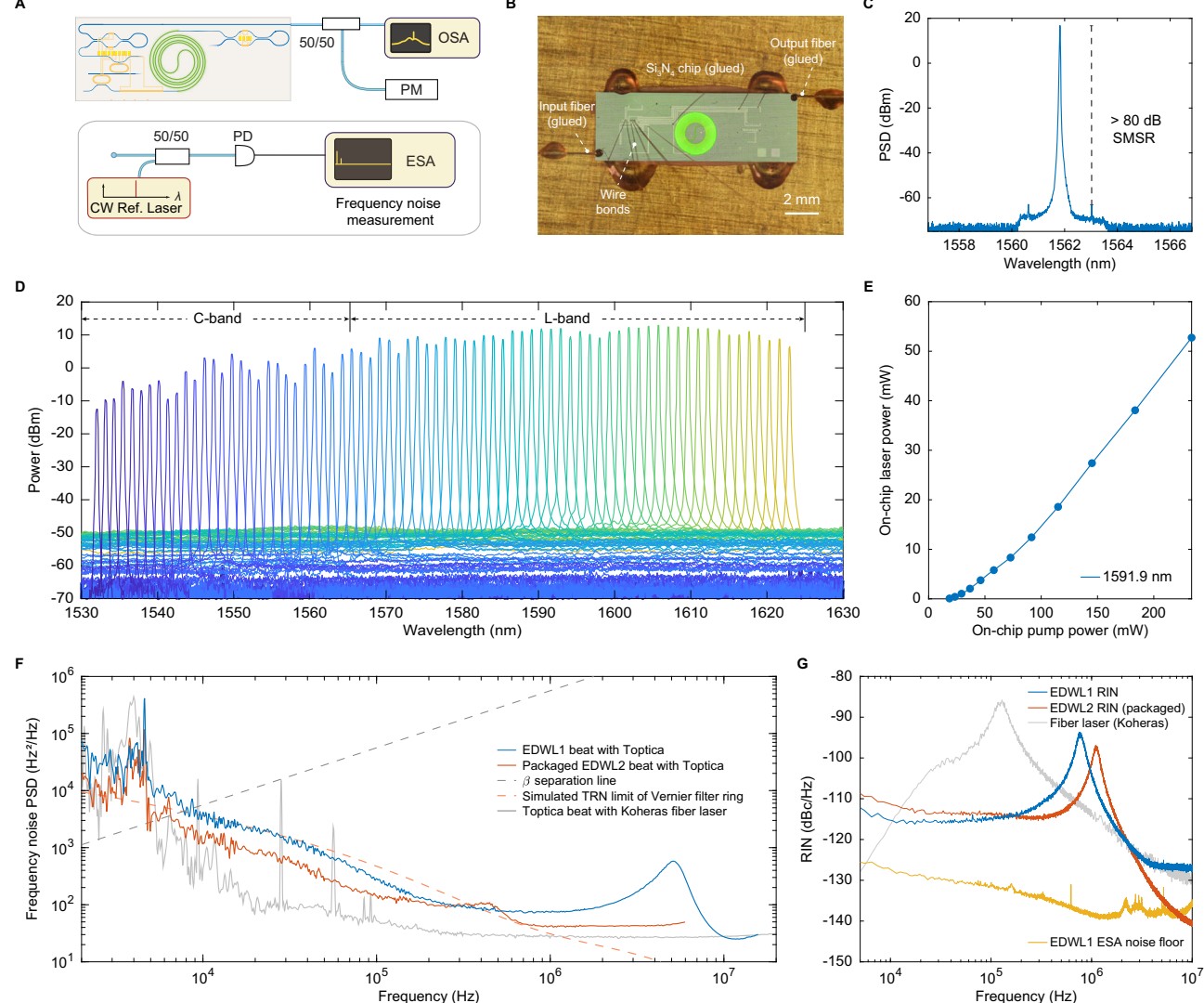

**Fig. 3 | Performance of EDWLs in optical C + L bands. A** Experimental setup for laser wavelength tuning demonstration. OSA: optical spectrum analyzer, PM: power meter, PD: photodiode, ESA: electric spectrum analyzer, CW Ref. laser - Toptica CTL. **B** Optical image of a fully packaged integrated Er-based tunable laser assembly. **C** Optical spectra of lasing with 47.6 mW output power measured in fiber, under remote pumping. The side mode suppression ratio (SMSR) is larger than 80 dB (Device ID: `D125_04_F2_C2`). The OSA resolution bandwidth (RBW) is set to 0.02 nm. **D** Optical spectrum of single-mode lasing across the optical C + L bands. Wavelength tuning is achieved by adjusting Vernier filter and loop mirror reflection with micro-heaters. Er ions are bidirectionally pumped using a fiber array with spacing matched to the separation between the input coupler and the mode-selective loop mirror, which transmits 1480 nm pump light and reflects the 1550 nm

lasing (Device ID: `D125_04_F9_C3`). The OSA RBW is 0.2 nm. **E** Dependence of lasing power on pump power, both measured on-chip, showing a 17.5 mW threshold and ≈24% slope efficiency at 1591.9 nm (Device ID: `D125_04_F4_C3`). **F** Laser frequency noise of free-running (blue) and packaged (red) EDWLs, using heterodyne detection with a reference ECDL laser. The frequency noise of the reference ECDL, measured via heterodyne detection with a stable fiber laser, is shown in grey. The β–separation line is shown in dashed grey. Simulated thermorefractive noise (TRN) limit for a single Vernier filter ring is shown in dashed red. EDWL 1 device ID: `D125_04_F4_C3`. EDWL 2 device ID: `D125_43_F2_C2`. **G** Measured RIN of EDWL1 (blue), EDWL2 (red), EDWL1 ESA noise floor (yellow), and comparison with a fiber laser RIN (Koheras).

reabsorption and lower passive loss. Further improvements are anticipated by reducing cavity and Vernier-filter insertion losses and mitigating Er-ion quenching.

We evaluated the single-sided power spectral density (PSD) of frequency noise by processing the in-phase and quadrature components of the sampled beatnote between the EDWL and an external cavity diode laser (ECDL, Toptica CTL) using Welch's method[41] (Fig. 3F). For a free-running EDWL lasing at 1592 nm (blue trace, EDWL 1), the frequency noise PSD beyond the offset frequency of 1 MHz exhibits a plateau of $h_0$ = 25 Hz²/Hz, corresponding to a Lorentzian linewidth $\pi h_0$ of 78.5 Hz. A full width at half-maximum (FWHM) linewidth of 99.7 kHz, associated with Gaussian contributions, was obtained by integrating the frequency noise from $1/T_0$ up to the

frequency where frequency noise PSD $S_{\delta\nu}(f)$ intersects the β-separation line $8\ln(2)\cdot f/\pi^2$ (dashed grey) at $T_0$=1 ms measurement time. A relaxation oscillation peak is observed at 5 MHz, calculated as $f_r \approx \frac{1}{2\pi}\sqrt{\frac{P_{cav}\kappa}{P_{sat}\tau}}$, where $P_{sat}$ is the saturation power of the gain medium, $P_{cav}$ is the laser cavity power, $\kappa$ is the cold cavity loss rate, and $\tau$ is the erbium ion upper-state lifetime. In the fully packaged laser, represented by the red trace (EDWL 2) measured at 1588 nm, a lower FWHM linewidth of 90.4 kHz is achieved by integrating over a 1 ms measurement time, attributed to reduced contributions from mechanical vibrations at lower offset frequencies. The relaxation oscillation frequency in EDWL 2 occurs at a lower offset of 0.5 MHz, primarily due to differences in laser cavity power. The frequency noise PSD of the reference ECDL (in grey) was obtained via heterodyne measurement

with a commercial fiber laser (Koheras ADJUSTIK) near 1550 nm. A similar PSD plateau $h_0$ is observed beyond 1 MHz, aligning with the characteristics of EDWL 1 and EDWL 2. Noise features below 5 kHz offsets are also comparable, indicating dominant contributions from the ECDL in this range for the measured EDWL PSDs. Using a reference laser with lower frequency noise beyond 1 MHz and below 10 kHz could further reduce the resultant Lorentzian linewidth $\pi h_0$ and FWHM linewidth of the EDWLs, demonstrating a performance comparable to or surpassing that of the ECDL in these frequency ranges.

Among the various factors influencing the EDWL's frequency stability, thermorefractive index noise (TRN) emerges as a significant contributor at offset frequencies below 1 MHz. Using established models[42,43], we computed the TRN for a single ring resonator in the Vernier filter and estimated its contribution to the overall laser cavity (detailed in Supplementary Note 7). The simulated TRN is shown as a dashed red line in Fig. 3F. Pump-induced intensity fluctuations can couple to EDWL frequency noise through thermal modulation of the cavity and Vernier rings[44]. To assess this effect, we measured the pump-to-frequency-noise transduction by applying a weak sinusoidal modulation to the 1480 nm pump and tracking the corresponding signatures in the pump RIN and EDWL frequency-noise spectra[45]. This transduction function is used to scale the measured pump RIN and estimate its contribution to the total frequency-noise spectrum (Supplementary Note 8).

The RIN of the EDWL, shown in Fig. 3G, was measured via photodetection using a DC-coupled photodetector (Thorlabs DET08CFC) with 0.9 A/W responsivity at 1550 nm. The measured EDWL RIN (blue) reaches values as low as −116 dBc/Hz at mid-range offset frequencies from 5 kHz to 1 MHz. A PSD pole is attributed to relaxation oscillation near 1 MHz. For reference, we measured the RIN of a fiber laser (Koheras) and observed a generally higher RIN than our EDWL, confirming that the EDWL achieves coherence comparable to, or surpassing, state-of-the-art fiber lasers.

The results in Fig. 3 are from identical EDWLs fabricated on the same wafer. Supplementary Note 9 compares devices across DUV stepper fields and shows similar tuning ranges and comparable frequency-noise levels at different wavelengths, confirming uniform wafer-scale reproducibility.

## Laser emission robustness against temperature and reflection

Semiconductor lasers prevail in industrial applications due to their compactness and maturity in mass production. They are highly sensitive to elevated temperatures, which lead to increased carrier recombination, higher threshold currents, reduced efficiency, and drifting of the gain center[46–48], necessitating improved thermal management and laser design[49]. In contrast, the erbium-doped gain medium is inherently temperature-insensitive due to its narrow, atomic-like optical transitions shielded by filled $5s$ and $5p$ electron shells, which suppress phonon interactions and stabilize erbium energy levels against temperature fluctuations[8]. As a result, erbium-doped gain media exhibit a minimal change in their emission cross-section and population inversion efficiency with temperature. This ensures that erbium-doped lasers maintain stable output power and linewidth, even at elevated temperatures, without the performance degradation commonly seen in semiconductor-based lasers.

Figure 4A presents the laser emission spectrum at 85 °C (6.3 mW), 105 °C (4.4 mW), and 125 °C (5 mW), demonstrating stable operation with a slight power reduction compared to room temperature (10 mW), primarily due to fiber-to-chip coupling drifts.

Another significant challenge faced by integrated lasers is back-reflections, which degrade coherence, introduce frequency noise, and cause severe nonlinear distortions for many analog and digital applications[50]. We characterized the stability of the Er-doped integrated laser under varying reflection conditions in Fig. 4. To evaluate the effect of reflections, we used the experimental setup shown in Fig. 4B, where reflection sensitivity was characterized by injecting controlled back-reflections into the laser cavity. Figure 4C, D present the spectrograms of the laser beatnote with external and self-reflections, respectively, monitored via delayed self-heterodyne detection. The RF beatnote, centered at 80 MHz, corresponds to the RF frequency applied to the acousto-optic modulator (AOM). The feedback levels were systematically varied from −30 to +10 dBm for external reflections. The laser demonstrated resilience to these reflections, maintaining a consistent frequency response without coherence collapse across the tested reflection levels. Notably, the laser maintained stability with self-reflections up to 14 dB attenuation, on par with, or better than conventional ECDLs such as the Toptica DL pro, which typically require >30 dB optical isolation to mitigate back-reflections. This robustness against back-reflections, attributed to the microresonator drop ports functioning as intrinsic bandpass filters in the EDWL, suggests potential isolator-free operation, simplifying photonic designs and enhancing scalability in demanding environments.

The detailed frequency noise characterization of the EDWL under external injection and self-reflection is presented in Supplementary Note 10. The measurements show that external injection, detuned by 217 MHz, does not measurably affect the DSH spectrogram or linewidth, confirming the robustness of the Vernier-filter design against back-reflections. In contrast, increasing self-reflection reduces low-offset frequency noise and narrows the integrated linewidth, consistent with coherent negative feedback via delayed self-injection (Lang–Kobayashi model[51–54]), particularly in devices with a small linewidth-enhancement factor $\alpha_H$, such as our EDWL.

Next, we characterized the long-term frequency stability of the integrated Er-doped Si$_3$N$_4$ laser (Fig. 4E) by monitoring the RF beatnote spectrum from its heterodyne interference with a fully-stabilized optical frequency comb (FC1500, Menlo Systems). The Er integrated laser is mounted in a 14-pin butterfly package with UHNA-7 fibers for Er pumping (Fig. 3B), while the III-V 1480 pump laser diode is positioned remotely to isolate heat-induced instability. The beatnote with center frequency of 250 MHz, derived from two comb lines of the optical frequency comb source, was used to monitor and benchmark stability. The laser-comb line beatnote was tracked over a 6-h period. The spectrum in Fig. 4E shows a frequency drift within 15 MHz, demonstrating strong stability in our Er:Si$_3$N$_4$ laser. This stability, attributed to the monolithic design with a high-$Q$ cavity and efficient thermal management, meets the stringent demands of advanced applications such as precision sensing, LiDAR, and high-coherence optical communications, where narrow linewidth, high stability, and disturbance resistance are critical.

## Discussion

In summary, we demonstrated the first EDWLs fabricated via wafer-scale processes, achieving an 80 dB SMSR and quasi-full C- and L-band tunability. Using 200-nm thick Si$_3$N$_4$ photonic integrated circuits, the implantation energy was reduced to below 500 keV, with scalability demonstrated through wafer-scale implantations in a commercial 300 mm tool, providing a credible path to high-yield industrial manufacturing. The laser delivers up to 47.6 mW fiber-coupled output power with a Lorentzian linewidth of 78.5 Hz and a frequency drift within 15 MHz over 6 h. A key advantage of the Er:Si$_3$N$_4$ Vernier laser architecture is the temperature insensitivity, enabling operation at up to 125 °C without significant performance degradation. The architecture also supports remote pumping, ensuring environmental stability and functionality in harsh conditions without requiring costly sealed packaging. Future improvements in gain coefficient and cavity loss could achieve fiber laser-level coherence, establishing the laser as a compact, mass-producible platform for next-generation optical systems, enabling applications in coherent sensing, LiDAR, analog optical links, and coherent communications.

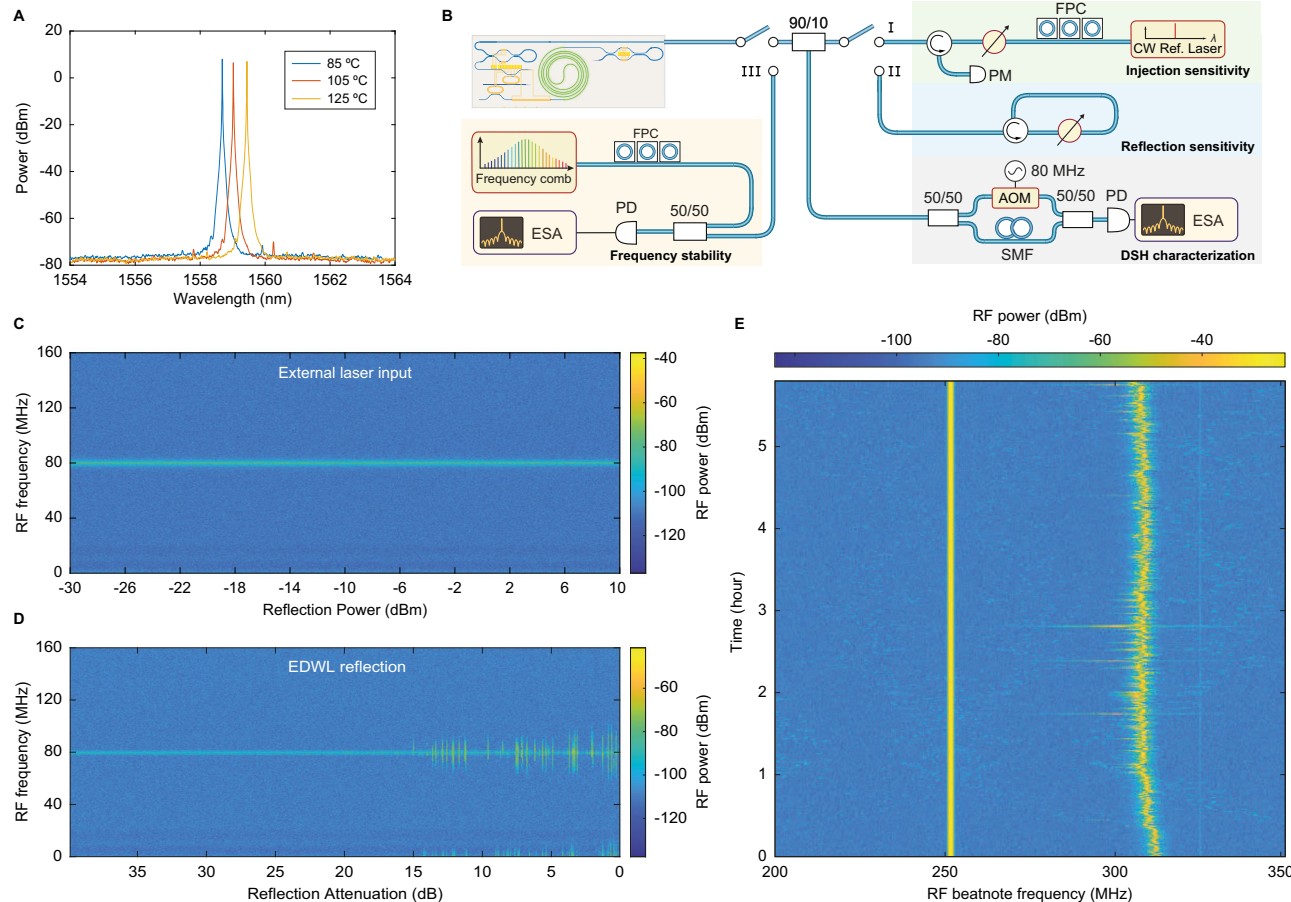

**Fig. 4 | Stability characterization of the Er-based integrated tunable laser.**
**A** Optical spectra of lasing at device temperatures from 85 °C to 125 °C, exceeding the temperature limit of most commercial III-V semiconductor lasers.
**B** Experimental setup for laser stability characterization, evaluating insensitivity to external reflections (Path I), self-reflections (Path II), and lasing wavelength drift through heterodyne beatnote measurement with a stable frequency comb (Path III). The RF beatnote is measured by a delayed self-heterodyne (DSH) interferometric measurement. The center frequency of 80 MHz is determined by the RF frequency applied to the acousto-optic modulator (AOM). SMF: single mode fiber delay line, FPC: fiber polarization controller. **C** Measured spectrogram of the laser

DSH beatnote when increasing the power of an injected external laser at the same wavelength, within a detuning of 217 MHz. **D** Measured spectrogram of the laser DSH beatnote under increasing self-reflection levels. The optical path with a circulator was modified to reflect the laser through a variable optical attenuator (VOA). **E** Measured spectrogram of the laser heterodyne beatnote with an optical reference system (FC1500, Menlo Systems) over 6 h, using remote pumping to mitigate thermal instabilities from direct integration. The peak at 250 MHz corresponds to the frequency comb repetition rate. The peak at 310 MHz is a beat of a comb line with the laser, drifting less than 15 MHz over 6 h.

## Methods

### Photonic integrated circuit fabrication

For the fabrication of passive, ultra-low loss $Si_3N_4$ photonic integrated circuits, a single-crystalline 100 mm Si wafer was wet-oxidized with 8.0 μm $SiO_2$ to provide a bottom cladding which can effectively isolate light confined in the thin $Si_3N_4$ waveguide from the silicon substrate, including at edge couplers. The gain-section waveguides, with a $5 \times 0.2$ μm² cross-section and a 3.02 μm² mode area (Fig. 1G), are formed from 200 nm thick LPCVD $Si_3N_4$ films. The LPCVD $Si_3N_4$ films exhibit an RMS roughness of 0.3 nm and uniformity of ±0.6% across a 4-inch wafer. Following deposition, $Si_3N_4$ films are annealed at 1200 °C for 11 h to eliminate excess $H_2$ and break N-H and Si-H bonds. This annealing process results in a 3.5% thickness reduction in the 200 nm $Si_3N_4$ layer due to film densification and hydrogen effusion[32]. In contrast to post-waveguide annealing in thick $Si_3N_4$ films[34], immediate annealing after LPCVD ensures precise thickness control and preserves the waveguide shape, preventing deformation caused by tensile stress in low-aspect-ratio waveguides (Supplementary Note 2). The waveguide patterns are defined using DUV stepper lithography (KrF 248 nm) with a resolution

of 180 nm. $Si_3N_4$ waveguides are formed via anisotropic dry etching with $CHF_3$ and $SF_6$ gases, yielding vertical, clean, and smooth sidewalls while minimizing polymer redeposition from etching byproducts.

After a fully wafer-scale fabrication process, the wafer was separated into dies of individual chip-scale lasers (Fig. 1D) via deep etching of $SiO_2$ and subsequent deep reactive ion etching of Si using the Bosch method, followed by backside grinding.

### Reporting summary

Further information on research design is available in the Nature Portfolio Reporting Summary linked to this article.

## Data availability

The experimental datasets used to produce the plots in this paper are available at Zenodo (https://doi.org/10.5281/zenodo.18459173).

## Code availability

The experimental scripts used to produce the plots in this paper are available at Zenodo (https://doi.org/10.5281/zenodo.18459173).

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

## Acknowledgements

We thank Professor Carsten Ronning for the Er photoluminescence measurements. The $Si_3N_4$ samples were fabricated in the EPFL Center of MicroNanoTechnology (CMi). This work has been supported by the Swiss National Science Foundation under grant agreement No. 211728 (BRIDGE), as well as by the European Research Council (ERC) under the Horizon Europe research and innovation programme, grant agreement No. 101167540 (ATHENS).

## Author contributions

Y.L. conceived the idea and concept. X.J. and Y.L. designed $Si_3N_4$ waveguide laser chips. X.J. and Z.Q. fabricated the $Si_3N_4$ samples. T.K. and J.C.O. performed the ion implantation on the $Si_3N_4$ samples. X.J, X.Y., Y.L. performed the experiments with support from G.L., S.B., Z.Q and J.S. X.J, Y.L., X.Y. carried out data analysis and simulations. A.V. designed and performed the device packaging. X.J. and Y.L. wrote the manuscript with input from all co-authors. T.J.K. and Y.L. supervised the project.

## Competing interests

T.J.K. is a cofounder and shareholder of LiGenTec SA, a start-up company offering $Si_3N_4$ photonic integrated circuits as a foundry service. T.J.K. is a cofounder and shareholder of EDWATEC SA, a start-up company offering optical amplifiers on chip. The remaining authors declare no competing interests.
