## [Transparent Peer Review file · Nature Communications]

Wafer-scale manufacturing of ultra-broadband, high-power erbium-doped integrated lasers

Corresponding Author: Professor Tobias Kippenberg

Version 1:

Reviewer comments:

Reviewer #1

(Remarks to the Author)
See the attachment.

[Editorial Note: See end of file]

Reviewer #2

(Remarks to the Author)

The authors present on wafer-scale Er-implanted SiN waveguide lasers with high powers, stable operation and wide tuning range. The main advances are implementing the lasers on a low confinement SiN platform (200 nm) with typical implant beam energies (< 500 keV), enabling mass production on 8 and 12 inch wafers. Although one might argue that the main overall ideas of Er ion implantation in SiN and Vernier lasing have been shown, the wafer scale integration of Er-implanted SiN lasers is a big step and it is combined with significantly improved performance. The paper is very well organized and the results are impressive and of general interest to the photonics community, and I recommend it be published in Nature Communications largely as it is. I just have some relatively minor comments:

- it would be helpful to add information on the SiN refractive index vs. Er doping concentration. The changes should be small but can impact the device design and possibly reflections in the laser cavity, and it is also important for simulating devices on the platform.
- I believe in Fig 1B Er:Al₂O₃ should be SiN/Er:Al₂O₃
- Figure 1 caption: part G - "The effective optical mode areas...are shown for comparison." Aren't these the ion implant energies?
- an exact absorption range of 1500-1538 nm is given related to hydrogen...this absorption can extend beyond those wavelengths, perhaps modify this slightly or give a reference that explains why this exact range for your material
- I suggest giving the FSRs and Vernier FSR in terms of wavelength in brackets as well for easy reference
- Indicate lasing wavelength for laser power measurement in Fig. 3E
- Perhaps it is a preference, but in general it might be preferred to have plot axes begin and end on specific labelled values for readability. Perhaps this is ok for the journal as is, but Fig. 4A and C, for example, particularly stand out as looking like they are missing an x axis label on the left
- in the conclusion, the first Er-doped waveguide laser fabricated by wafer scale processes is claimed - this should be refined a bit since others have demonstrated wafer-scale on-chip Er lasers. E.g. first Er-implanted SiN wafer scale laser. In fact the statement in the abstract should also be refined a bit, since foundry compatible tunable Er waveguide lasers have been shown in the past.
- the Methods section seems a bit unnecessary, since it is very short and fabrication details also appear in the main body and supplementary material. I suggest condensing this into two places instead of three and eliminating the Methods.
- Supplementary material: The supplementary material is very helpful and thorough. I just have a few suggestions:
-> Fig S5: indicate that it is the cladding or SiO₂ in the legend, otherwise it might be mistaken for H in the SiN (which is a typical consideration)

- > Section 4: perhaps I'm missing it, but in case not, give the cavity roundtrip length value
- > elaborate on what is meant by "wavelength-independent passive loss"
- > the following is somewhat confusing: "The L-band EDWL output power matches the simulated value, indicating a smaller O-H absorption tail in real devices than initially modeled."
- > I strongly suggest including an experimental diagram/diagrams illustrating the setup and how the measurement is done for section 5 (e.g. in Figure S7) since it is a bit hard to follow
- > section 6 can use a few more sentences explanation explaining defining thermal refractive noise, perhaps a reference such as the authors' own work and how it manifests in resonators and how temperature plays a role at the beginning, since temperature doesn't appear in the math
- > Section 7: add setup schematic for heterodyne interferometry or refer to main text if it is there

- it would be good to check for grammar/typos, I spotted the following:

- > a 8 μm -> an 8 μm
- > on ultra-low loss Si₃N₄ PICs fabrication -> on the ultra-low loss Si₃N₄ PICs fabrication
- > supplementary material: normal -> nominal
- > supplementary material: in EDWL -> in the EDWL or in EDWLs

Reviewer #3

(Remarks to the Author)

The authors have demonstrated an integrated design and operation of Er-ion doped Si-nitride based tuneable laser device operating in the C+L band. The temperature tuneability and noise figures see quite impressive. However, there are several comments which might help the authors to improve the quality of paper and also set the scene for future development.

a) The main improvement in the Si₃N₄-based waveguide is the quality of deposited films and low-OH loss in silica. When silica mixes with Si₃N₄, it forms silicon oxynitride and the OH-ions therein at the interface will overlap with Er-ion absorption band, and thereby contribute to OH-concentration quenching. There is no data to show how much reduction in OH was achieved as a result of the process control. Was it the high 2.0MeV the cause of larger OH-absorption or, something else. This is unclear.

b) Did the structure of Si₃N₄ change when the implantation energy was dropped from 2.0MeV to 0.5MeV. The higher implantation energy will also allow Si₃N₄ to implant deeper in the silica layer and contribute to OH-induced loss.

c) The authors demonstrate spontaneous emission spectrum of the Si₃N₄/SiO₂ film in Figure 2B is measured with an Argon-ion laser, which is usually at 514nm and not at 520nm? This excitation wavelength also shows the emission features in the visible and near-IR which are important to include.

d) In the context of this paper, the authors have used the 1480nm laser which also overlaps with the Er-ion and potentially with the 1st harmonic of the OH-ion. This data is missing from the article which the authors recognise but the quantifiable data are not included (in the article or supplementary section). The loss of gain at shorter wavelength may be due to OH-quenching and Er-ion absorption and re-emission at longer wavelengths (See comment e).

e) The authors have referred Profs. Desurvire and Zervas text book. (E. Desurvire and M. N. Zervas, Erbium-doped fiber) so I presume that that the original reference (Desurvire, Giles, Simpson and Zyskind) in Optics Letters. The articles published by Desurvire et emphasize the importance of Er- population redistribution due to temperature effect and, therefore, the tuneability. My question is with the temperature control the tuneability might be achieved? would this than also require higher energy to maintain and cool the device to achieve broader tuning range. Also, downstream the amplifier might also have a similar issue?

f) The key question for me is to understand whether there might be spectral hole creation in the ground and 4I_{13/2} states which will be dependent on temperature, so a say 1550.15nm signal might be absorbed and become a longer wavelength signal. The Optics Letter of Desurvire et al discusses the transitions.

g) There is a mistake in the paper with reference to 4f electronic level shielding? The Er-ion 4f electrons are shielded by 6s and 5d and NOT 5s and 5p? Please read the article and correct.

Version 2:

Reviewer comments:

Reviewer #1

(Remarks to the Author)

Reviewer #3

(Remarks to the Author)

Thank you for answering all the relevant questions raised by Reviewer 3. The queries have been resolved and incorporated

in the article and supplementary information. It would be quite nice to include a high-resolution SEM image with element mapping across the Silica/Silicon Nitride interface. The high-resolution elemental map might be able to show the sharpness of interface and potential changes in the refractive index.

Submission of a revised version of manuscript NCOMMS-25-17303A-Z

Wafer-scale manufacturing of ultra-broadband, high-power erbium-doped integrated lasers

Dear Editor,
Dear Reviewers,

We are grateful that our manuscript has been seen by three reviewers. We appreciate that the reviewers read it carefully, made suggestions and gave an overall positive evaluation. In the following, we present a point-by-point reply (in **blue**) to the reviewers' comments (in **black**), as well as the action taken (in **red**). References are cited as '[X]' for those in the main text, '[SX]' in the Supplementary Information, and '[RX]' in this response letter.

In the revised manuscript, according to the reviewers' request, **we have updated the main text with additional details and measurements and added new figures to the Supplementary Information**. The main changes are:

- (i) We added wafer-scale lasing and frequency-noise measurements.
- (ii) We updated the simulations of output power, slope efficiency, and fundamental linewidth using a travelling-wave rate-equation/propagation model.
- (iii) We revised the explanation for reduced C-band output.
- (iv) We report and analyze frequency-noise reduction with increased self-reflection.
- (v) We also expanded Supplementary Sections 7 and 8, providing clearer derivations for pump-RIN-to-frequency-noise transduction and for thermal-refractive noise in Vernier ring resonators.

Reviewer #1

The work on integrated photonics with rare-earth is quite interesting in silicon nitride, and the authors here are leading this effort. They have presented already their works in several high impact journal (Y. Liu et. al. A fully hybrid integrated erbium-based laser, Nature Photonics, 18 2024, Y. Liu et. al. A photonic integrated circuit-based erbium-doped amplifier, Science, 376 (2022)). The current work is quite similar to these previous works and the performance level, foot print etc. and the main claim wafer level demonstration is not properly shown - in terms of yield and chip to chip performance over the entire wafer and from wafer to wafer. See my comments below for detail.

We thank the reviewer for the positive assessment of our previous work and for the interest in rare-earth ion implantation in silicon nitride.

1) Authors say in the abstract "The reduced implantation energy marks a crucial advance toward scalable production of Er-doped photonic devices. Meanwhile, the increased optical mode area of low-confinement Si₃N₄ waveguides significantly enhances laser performance and output power." Could you please explain why it is not considered obvious that a "thin" film silicon nitride will require less energy implantation. Also increased mode area has been explored recently in integrated photonics for high power output which has not been cited in the manuscript it seems – N. Singh. Light Science Appl. 14, 2025, N. Singh et. al. Nat. Photonics, 19, 2025 etc. But the geometry seems different here so there is some novelty but not entirely new in integrated photonics.

We agree it is obvious that a thinner Si₃N₄ core requires lower implantation energy. Our claim of a "crucial advance" refers to the practical step-change this enables: by reducing the required energy to ≤ 500 keV, erbium

implantation falls within the capability of standard, high-throughput CMOS implanters, yielding minutes-scale wafer processing rather than hours on specialized MeV beamlines limited to chip-scale fields. This materially improves manufacturability and cost. Figure 1G quantifies the energy needed to reach the waveguide center versus thickness, and Figure 1H benchmarks these energies against microelectronics implant parameters, underscoring compatibility with wafer-scale, CMOS-compatible production. This is the basis for our statement that reduced implantation energy “marks a crucial advance toward scalable production of Er-doped photonic devices.”

Regarding mode area, we will cite the recent large-mode-area (LMA) work by N. Singh et al. (Light: Science & Applications, 2025; Nature Photonics, 2025). Those studies employ LMA geometries (mode areas of order tens of μm^2) to raise saturation power and mitigate nonlinearities in amplifiers and lasers; LMA waveguides/lasers are realized by sputtering Tm:Al₂O₃ on Si₃N₄ to form hybrid modes. In contrast, our approach is complementary: we increase mode area by low-confinement Si₃N₄ geometry and incorporate gain by direct ion implantation into the Si₃N₄ core, enabling spatially selective doping with high mode overlap while preserving low loss in passive sections (e.g., Vernier rings and loop mirrors protected during implant). Unlike LMA transitions, our method avoids large bends and mode-transition sections and delivers wide tunability and high coherence from a monolithic cavity. We added the Singh et al. citations and clarify these distinctions in the revised manuscript.

Action taken:

We cite the LMA literature and contrast it with our architecture:

“Prior wafer-scale rare-earth lasers deposited Er:Al₂O₃ on Si or Si₃N₄ waveguides [20, 21], and mode-hybrid Tm:Al₂O₃/Si₃N₄ large mode area (LMA) amplifiers and lasers have scaled on-chip power [22, 23], but neither is a monolithic, directly implanted Er:Si₃N₄ platform. Here we directly implant Er into Si₃N₄ waveguides and demonstrate wafer-scale, C+L-band tunable waveguide lasers with fiber-laser-class coherence without external seeding [24].”

[20] Purnawirman, et al. C-and L-band erbium-doped waveguide lasers with wafer-scale silicon nitride cavities. *Optics letters* 38.11: 1760-1762 (2013).

[21] N. Li, et al. Monolithically integrated erbium-doped tunable laser on a CMOS-compatible silicon photonics platform. *Optics Express* 26.13: 16200-16211 (2018).

[22] N. Singh, et al. Towards CW modelocked laser on chip—a large mode area and NLI for stretched pulse mode locking. *Optics express* 28.15, 22562-22579 (2020).

[23] N. Singh, et al. Watt-class silicon photonics-based optical high-power amplifier. *Nature Photonics* 19.3, 307-314 (2025).

[24] N. Singh, et al. Sub-2W tunable laser based on silicon photonics power amplifier. *Light: Science & Applications* 14.1,18 (2025).

2) In the introduction authors say the noise figure is low for Er but that is usually true for fiber, however it’s not obvious if the noise figure of Er waveguide amplifier is substantially better than semiconductor amplifiers which are additionally electrically pumped.

We thank the reviewer for the comment on noise figure of EDWA. Our analysis and measurements indicate that EDWAs retain the EDFA-class low NF and remain substantially below SOAs: In SOAs, interband gain with carrier-induced index changes leads to strong amplitude-phase coupling and fast (sub ns) carrier dynamics. Consequently, the effective spontaneous-emission factor n_{sp} and internal loss are higher than in erbium amplifiers, giving intrinsically larger noise figures. In contrast, erbium gain arises from an atomic, parity-forbidden intra-4f transition ($^4I_{15/2}$ - $^4I_{13/2}$), characterized by slow dynamics, negligible inter-channel cross-talk, high-temperature stability and low noise figure. In our Si₃N₄-based EDWAs, we observe a luminescence spectrum similar to that in silica (Fig. 2B), with an excited-state lifetime of 3.4 ms (Supplementary information section 4). These characteristics confirm that EDWAs share the same fundamental gain mechanism as EDFAs.

When the waveguide loss is low and the pump drives the medium close to full inversion ($n_{sp} \rightarrow 1$), the EDWA noise figure likewise falls within the EDFA class, typically around 3-6 dB.

To validate this, we measured the NF of a stand-alone 17 cm Er:Si₃N₄ EDWA (same design as used in the EDWL) using the standard optical source-subtraction method [R1], which removes source-spontaneous-emission (SSE) from the measured noise:

$$NF = \frac{P_{out} - GP_{SSE}}{Gh\nu B_0} + \frac{1}{G}$$

Figure R1: Measured noise figure of the 17-cm-long Er:Si₃N₄ waveguide amplifier under various off-chip input powers at 1550 nm.

Pump light is launched bidirectionally with 330 mW off-chip power on each side. The measured fiber-to-fiber insertion loss is 2.5 dB. Figure R1 reports the EDWA's NF versus off-chip signal input power: for weak inputs (high-gain regime) we obtain NF ~ 4-5 dB; as the input increases and the amplifier saturates, the NF rises. These values include fiber-to-chip coupling and other insertion losses, reducing coupling loss will move the effective NF toward the intrinsic erbium range. The measurements in Fig. R1 were performed at 1550 nm; at longer wavelengths the NF is expected to decrease due to weaker signal reabsorption [R2].

We further consider the impact of classical pump diode noise. In principle, pump power fluctuation can modulate inversion and gain. However, EDFAs are not sensitive to high-frequency power fluctuations because the upper excited state level is typically long-lived (10 ms level lifetime, 3.4 ms for EDWA) and serves as an energy reservoir. Consequently, pump power fluctuations at frequencies above 1 kHz can hardly influence amplified signals. Only low-frequency fluctuations can impact the signal, but these are usually quite weak (we measure RIN of -120 to -130 dBc/Hz for our 1480 pump laser in Figure S15(b) (updated measurement)). We note that EDFAs are likewise optically pumped by electrically driven diodes, thus the NF differences arise from the absorption and emission of Er ions, and internal loss and reflections, not from pump type. In the EDWL, pump-induced gain fluctuations can convert to output RIN, which we discuss separately in supplementary information section 5; this does not contradict the low NF of the amplifier.

[R1] D. Derickson, et al. Fiber optic test and measurement. *Fiber optic test and measurement*/edited by Dennis Derickson (1998).

[R2] E. Desurvire, et al. Analysis of noise figure spectral distribution in erbium doped fiber amplifiers pumped near 980 and 1480 nm. *Applied Optics* 29.21: 3118-3125 (1990).

Action taken:

We discuss the noise figure and include the noise figure measurement in the Supplementary information:

“We measured the noise figure (NF) of a stand-alone 17 cm Er:Si₃N₄ EDWA (same cross-section as used in the EDWL) using the standard source-subtraction method [S27] at 1550 nm under bidirectional 1480 nm pumping, with 330 mW off-chip power on each side. The fiber-to-fiber insertion loss was 2.5 dB. The measurement results are presented in Fig. S21. In the high-gain regime (weak inputs), the system NF is approximately 4-5 dB; as the input increases and the amplifier saturates, the NF rises. These values include

coupling and passive losses; reducing those losses and operating closer to full inversion ($n_{sp} \rightarrow 1$) moves the intrinsic EDWA NF into the EDFA class. Because signal reabsorption decreases toward longer wavelengths, the NF is expected to be lower in the L-band [S28]. The long erbium upper-state lifetime ($\tau_{21} \approx 3.4$ ms) passively filters pump fluctuations, so only low-frequency pump noise can appreciably modulate gain; our pump RIN (-120 to -130 dBc/Hz, Fig. S15(b)) makes this effect negligible in the NF context [S3]. For comparison, semiconductor optical amplifiers typically exhibit larger effective n_{sp} and internal loss due to interband gain with fast carrier dynamics, leading to higher NF than erbium-doped amplifiers.”

[S27] D. Derickson, et al. *Fiber optic test and measurement*. Fiber optic test and measurement/edited by Dennis Derickson (1998).

[S28] E. Desurvire, et al. *Analysis of noise figure spectral distribution in erbium doped fiber amplifiers pumped near 980 and 1480 nm*. *Applied Optics* 29.21: 3118-3125 (1990).

[S3] Y. Liu, et al. *A photonic integrated circuit-based erbium-doped amplifier*. *Science* 376.6599: 1309-1313 (2022).

3) What is the variation in performance from chip to chip in the wafer? It would be useful to know, if the claim is about wafer level performance, the variation and yield for data shown in this work, power, tunability, linewidth etc. Such data is quite important for wafer lever demonstration.

We thank the reviewer for the question regarding wafer-scale performance variation. We evaluated EDWLs from the nine central stepper fields (Fig. R2), measuring their tuning spectra and frequency noise under unidirectional pumping (~ 380 mW off-chip).

Figure R2: **DUV stepper exposure layout**: a 5×5 array of 2.024 cm \times 2.024 cm fields; the nine central fields were fully exposed and used for device characterization.

Across fields, the C+L-band tuning spectra are qualitatively consistent, indicating reproducible wafer-scale fabrication. The observed span variations arise primarily from $\sim 1.2\%$ LPCVD Si_3N_4 thickness nonuniformity across the 4-inch wafer, which perturbs Vernier ring FSRs and the achievable loop-mirror reflectivity range; fiber-to-chip coupling drift further varies on-chip pump power that affect the tuning range.

Figure R3(j)-(k) shows EDWL frequency noise measured by heterodyne detection against a low-noise local oscillator (Toptica ECDL): for a representative device (field F7, Fig. R3(j)), spectra from 1548-1619 nm show similar behavior at low-offset frequencies, with a common intrinsic floor $S_{\delta\nu} < 100$ Hz²/Hz above 1 MHz, while the relaxation-oscillation peak shifts slightly with output power. At 1559 nm, frequency noise across all nine fields shows no systematic dependence on field location. These results support uniform tunability and frequency-noise performance across the wafer. Potential improvements include implementing bidirectional pumping and redesigning the Vernier filter for full C+L-band coverage; tightening LPCVD film-thickness control (currently we can achieve a thickness uniformity of 0.6%); and adopting a more fabrication-tolerant mirror (e.g., a looped, imbalanced MZI).

Action taken:

We include the wafer-scale tuning spectra and frequency-noise measurements in a new Supplementary Information section, “**Wafer-scale uniformity of EDWL tunability and frequency noise**”:

“We evaluated wafer-level reproducibility by testing EDWLs from nine stepper fields (F1–F9) on a 4-inch wafer. The lithography layout is a 5×5 array of 2.024 cm×2.024 cm fields; the nine central fields were fully exposed and selected for characterization. Each EDWL was tuned by heating the Vernier microrings and was unidirectionally pumped at 1480 nm with ~380 mW off-chip power.

Figure S16 summarizes the results: Fig. S16(a)–(i) show C+L-band tuning spectra that are qualitatively consistent across all fields, indicating reproducible wafer-scale fabrication and high functional yield. Variations in span are attributed primarily to ~1.2% Si₃N₄ thickness non-uniformity across the wafer, which perturbs the Vernier-ring FSRs and the attainable loop-mirror reflectivity range. Additional variation arises from fiber-to-chip coupling drift that changes the on-chip pump power. Implementing bidirectional pumping, tightening LPCVD Si₃N₄ thickness control (<0.6%), and adopting a more fabrication-tolerant mirror (e.g., a looped MZI with intentional arm imbalance) would further equalize the tuning span.

Fig. S16(j) reports EDWL frequency-noise spectra (heterodyne measurement with a low-noise Toptica ECDL) for a representative device in field F7 measured from 1548–1619 nm. The spectra exhibit comparable low-offset behavior and a common intrinsic floor $S_{\delta\nu} < 100 \text{ Hz}^2/\text{Hz}$ above 1 MHz. The relaxation-oscillation peak shifts slightly with instantaneous output power. Fig. S16(k) shows frequency noise at 1559 nm for devices from F1–F9, revealing no systematic dependence on field location. Collectively, these results demonstrate wafer-scale reproducibility of EDWL performance.”

Figure R3: **Wafer-scale characterization of the EDWL tunability and frequency noise.** (a)–(i) Tuning spectra for EDWLs from stepper fields F1–F9 with unidirectional pumping (~380 mW off-chip). (j) Frequency-noise spectra of the EDWL in F7 at 1548–1619 nm, showing comparable low-offset noise and a similar intrinsic noise floor; relaxation-oscillation peaks shift with output-power variation. (k) Frequency-noise spectra at 1559 nm for devices from F1–F9.

4) And is the implantation over the entire wafer? If so then doesn't it affect the devices which are passives and co-integrated, for example the loop mirrors and rings shown in this work? How is their loss affected? Seems like this will increase the loss quite a lot and impact other devices.

Erbium is implanted as a blanket step to enable uniform, wafer-scale doping, using a standard semiconductor implanter at *Applied Materials* (Fig. 1H). Passive components are not implanted: they are protected by a $\sim 6 \mu\text{m}$ photoresist mask during the implantation (Fig. 1A and Methods), so only designated waveguide sections are exposed. This masking ensures that passive components such as loop mirrors and ring resonators remain undoped and retain their low-loss performance.

Following implantation, the wafer undergoes a 1000°C annealing to activate the erbium ions and reduce implantation-induced damage in the doped waveguides. This thermal process does not adversely affect the masked passive regions. Measurements of passive components on the same wafer confirm that performance is preserved: Si_3N_4 microresonators exhibit 2-4 dB/m propagation loss and the loop mirrors show broadband, low-loss transmission (Fig. 2D-G). Thus, co-integrated passive devices remain undoped and low-loss, and wafer-scale implantation does not impose a penalty on their performance.

5) Also the device foot print is quite large (40 mm^2) compared to their previous work 6 mm^2 (Y. Liu et. al. *Nature Photonics* 2024).

We thank the reviewer for noting the footprint difference from our prior work (Y. Liu *et al.*, *Nature Photonics*, 2024). The larger area ($\sim 40 \text{ mm}^2$) in this study is a consequence of moving to $200 \text{ nm Si}_3\text{N}_4$ (vs. 700 nm previously) to enable wafer-scale Er implantation at reduced energies compatible with standard semiconductor implanters. The thinner core lowers mode confinement, requiring larger bend radii to suppress radiation loss and wider spiral gaps to mitigate intermodal coupling; the design therefore prioritizes low loss and robustness over minimum footprint.

The present layout, however, is not footprint optimized. In the Supplementary Information, we include an alternative layout of the same cavity architecture with an area of $\sim 23.8 \text{ mm}^2$ (Fig. R4), reducing the footprint by nearly one half and thereby doubling the potential wafer yield without degrading performance:

Figure R4: Alternative EDWL layout with a reduced footprint of 23.8 mm^2 .

Eventually, as the one of the loop mirrors is operated at high reflectivity to reduce cavity loss, it can be replaced with a broadband waveguide Bragg grating, which provides near-unity reflection in a more compact form (Fig. R5), though at the cost of reduced tuning range due to its finite stopband:

Figure R5: Alternative EDWL layout using a waveguide Bragg grating (WBG) as one cavity reflector. The footprint is reduced to $\sim 19.52 \text{ mm}^2$. Inset: proposed WBG design.

Action taken:

We add a Supplementary Information section “**Footprint considerations and compact layout**”, presenting alternative, smaller-footprint layouts:

“The large area of our EDWL ($\sim 40 \text{ mm}^2$) reflects design choices of the $200 \text{ nm Si}_3\text{N}_4$ platform used to enable wafer-scale erbium implantation at reduced, standard-implanter-compatible energies. The thinner waveguide core reduces mode confinement, necessitating larger bend radii to suppress radiation loss and wider spiral gaps to mitigate intermodal coupling. The layout in Figure 1D thus prioritizes low propagation loss and robustness over minimum footprint.

To illustrate footprint scalability without altering the cavity architecture, we designed an alternative compact layout with an area of $\sim 23.8 \text{ mm}^2$ (Fig. S22(a)). This design preserves the Vernier-filtered cavity, gain-spiral length, and coupling conditions, and is expected to maintain comparable lasing performance while nearly halving the footprint, implying roughly a twofold increase in yield on a given wafer.

As one of the loop mirrors is operated at high reflectivity to minimize cavity loss, it may be replaced by a broadband waveguide Bragg grating, which provides near-unity reflection in a more compact form (Fig. S22(b)) at the expense of a tuning range limited by its finite stopband.”

Figure S22: **Compact layouts of EDWLs.** (a) Schematic of an alternative floorplan ($\sim 23.8 \text{ mm}^2$) that preserves the cavity architecture and gain length while reducing bend count and routing area; (b) a broadband waveguide Bragg grating can replace the WDM loop mirror to further compress footprint.

6) Also this work has slightly improved tuning range compared to their previous work (Y. Liu et. al. Nature Photonics 2024), but what is not shown is how this improvement is from device to device, chip to chip on the

entire wafer, essentially what is the yield? This is a key metric if the main claim is about wafer level. As semiconductor amplifiers are manufactured at industrial scale with high yield which this work compares to.

We respectfully disagree with the statement that “this work has *slightly* improved tuning range compared to previous work.” The present EDWL spans 91 nm (1530-1621 nm), which spans nearly the entire C+L band, and is 2.45 times broader than the 37 nm range reported in our previous work (1548-1585 nm, Y. Liu *et al.*, *Nature Photonics*, 2024). In addition to the broader tuning range, the present work increases the maximum fiber-coupled output power from 17 mW to 47.6 mW (updated Fig. 3C).

Regarding yield, as detailed in Q3, devices from nine stepper fields exhibit consistent tuning spectra and frequency noise, demonstrating wafer-scale reproducibility. Our process, using standard implanters and blanket implantation with photoresist masking, provides a reliable path to high-yield manufacturing. Achieving industrial-scale metrics will depend on routine process-control refinements such as implantation dose/energy uniformity and across-wafer thickness control.

Action taken:

We comment on the potential for industrial-scale manufacturing of our EDWL:

“Using 200-nm thick Si_3N_4 photonic integrated circuits, the implantation energy was reduced to below 500 keV, with scalability demonstrated through wafer-scale implantations in a commercial 300 mm tool, providing a credible path to high-yield industrial manufacturing.”

7) What is the measured coupling loss? Simulation value is only mentioned.

Figure R6: Insertion loss (fiber-to-fiber) through a reference waveguide with UHNA-7 fibers.

Figure R7: Simulated coupling efficiency for the edge coupler used in the EDWL. (a) Coupling efficiency from a 203 THz Seminex DFB pump laser to the chip as a function of the edge coupler width and height. (b) Coupling efficiency from UHNA 7 fiber to the chip. In both plots, the red star indicates the chosen edge coupler geometry.

We now report measurements alongside simulations. For a test waveguide with 0.5 μm wide, 200 nm thick edge couplers, the measured fiber-to-fiber insertion loss at 1550 nm is 2.566 dB (Fig. R6), i.e., ~ 1.28 dB per facet. Across 1520-1630 nm, the coupling ranges 1.24-1.63 dB per facet (2.48-3.26 dB fiber-to-fiber). Simulations for the EDWL coupler predict ~ 0.56 dB per facet at 1550 nm (Fig. R7(b)). The remaining gap is attributed to fabrication-induced taper width/height deviations (including tip erosion) and small alignment errors that increase overlap loss. Further reducing the UHNA-7 to SMF-28 splice loss should also narrow this gap.

Action taken:

We add a new Supplementary section, “**Edge coupler characterization**”, reporting calibrated fiber-to-fiber transmission measurements:

“Insertion loss was measured on a reference waveguide with 0.5- μm -wide, 200-nm-thick edge couplers identical to the output edge coupler of our EDWL. In Figure S5, the measured fiber-to-fiber insertion loss at 1550 nm is 2.566 dB (1.28 dB per facet), varying between 2.48 dB and 3.26 dB over the 1520-1630 nm wavelength range. Simulations for the EDWL coupler predict ~ 0.56 dB per facet at 1550 nm. The discrepancy is attributed to fabrication-induced taper width/height deviations (including tip erosion) and small alignment errors that increase overlap loss. Further reducing the UHNA-7 to SMF-28 splice loss should also narrow this gap.”

8) The loss mentioned on page 3 is from 1.5 to 4 dB/m. Is that before ion implantation or after, and what is the loss before and after annealing?

The 1.5 to 4 dB/m loss reported in Fig. 2D was measured after the complete process flow, i.e., following erbium ion implantation, 1000 $^{\circ}\text{C}$ activation anneal, SiO_2 cladding deposition, and chip singulation. The value is extracted from passive (undoped) LPCVD Si_3N_4 ring resonators (5 $\mu\text{m} \times 200$ nm cross-section, 50 GHz FSR) fabricated on the same wafer as the EDWL and protected by a 6 μm photoresist mask during implantation. Consequently, these resonators remain undoped and report the baseline passive platform loss after all post-implant steps have been applied.

Two annealing steps are used: a post-deposition Si_3N_4 anneal and a post-implantation Er activation anneal. As is well known for LPCVD Si_3N_4 , hydrogen-related overtones from Si-H and N-H bonds produce absorption features in the 195-199 THz band. We perform a 1200 $^{\circ}\text{C}$, 11 h anneal (prior to implantation, detailed in Methods and Supplementary section 2) to remove hydrogen-related bonds and densify the Si_3N_4 film, thereby establishing a low-loss baseline. Figure R8 illustrates how thermal annealing of Si_3N_4 affects a 50 GHz FSR resonator, indicating that sufficient annealing is required to achieve ultra-low loss. After this high-temperature step, the subsequent 1000 $^{\circ}\text{C}$ Er activation anneal (performed at wafer-scale after implantation) does not measurably change the passive resonator loss: the dominant hydrogen-related absorption has already been mitigated by the 1200 $^{\circ}\text{C}$ treatment, and the additional thermal process does not degrade the Si_3N_4 or SiO_2 .

Figure R8: Impact of thermal annealing conditions on the loss of Si_3N_4 waveguides. (a) Resonator loss processed with 6 hours annealing for Si_3N_4 at 1200 $^{\circ}\text{C}$. (d) Resonator loss processed with 11 hours annealing for Si_3N_4 at 1200 $^{\circ}\text{C}$.

In the doped waveguides, the 1000 °C activation anneal reduces background (out of Er absorption band) loss relative to the post-implant state by healing implantation damage (e.g., vacancy/defect annihilation) and promoting beneficial redistribution of implanted ions: In Fig. S1, the loss of doped resonators outside the Er absorption band is comparable to that in Fig. 2D (passive rings), confirming that the activation anneal restores low background loss while leaving the passive Si₃N₄ platform loss unchanged. This is consistent with our prior study [R3].

[R3] Y. Liu, et al. A photonic integrated circuit–based erbium-doped amplifier. *Science* 376.6599: 1309-1313 (2022).

Action taken:

We clarify in the manuscript that the reported 1.5-4 dB/m loss is measured after the complete fabrication flow:

“Figure 2D presents the TE-mode propagation loss of a passive microring resonator with a 50 GHz FSR and 5 μm width, measured by scanning laser spectroscopy after the complete process flow, reflecting the baseline passive platform loss.”

9) Are all the measurements in Fig.3 for different devices with only best performing numbers shown? It seems that is the case as device IDs are different for all. Please mentioned that in the manuscript, because then it seems that the high power device with 40 mW on chip power is not the best for wide tunability etc. This seems to suggest the yield and performance is not comparable to wafer level semiconductor devices.

We thank the reviewer for the careful reading. Figure 3 compiles results from several EDWLs fabricated on the same wafer with an identical mask, cavity architecture (waveguide cross-section, Vernier filter, loop mirrors), and process flow. Only the stepper field (device ID) differs. This was intentional to demonstrate wafer-scale operation. Additionally, our wafer-level study in Q3 shows that devices from nine stepper fields exhibit similar C+L-band tuning and frequency-noise spectra without systematic field dependence, and the measured frequency-noise levels are comparable to those in Fig. 3F. The widest-tuning device in Fig. 3D was operated at a lower on-chip pump than the high-power device in Fig. 3C; under comparable pump and coupling, similar output power is expected.

Action taken:

We will cross-reference the wafer-scale tunability and frequency-noise data in the SI to document reproducibility. These additions clarify that the results reflect wafer-level performance rather than isolated “best-case” devices:

“The results in Figure 3 are from identical EDWLs fabricated on the same wafer. Supplementary Note 9 compares devices across DUV stepper fields and shows similar tuning ranges and comparable frequency-noise levels at different wavelengths, confirming uniform wafer-scale reproducibility.”

10) Is the laser mode hop free over the entire bandwidth? And the laser seems to be better at 1608 nm rather than at the peak of erbium gain, why is that?

In the strict sense of “mode-hop-free,” meaning continuous tuning of a single longitudinal mode, our EDWL does not operate in this regime. Instead, wavelength tuning is achieved by thermally shifting the Vernier ring resonances to select successive cavity modes at different frequencies. Practically, however, the laser is single frequency at every setpoint and shows no uncontrolled mode hopping at increasing pump power (Fig. 3E). Under fixed heater bias the lasing mode is stable, and coordinated tuning of the two rings yields deterministic and repeatable lasing across the C+L band. Continuous tuning of the lasing wavelength can be achieved by simultaneous adjustment of the Vernier ring heater powers [R4]. In conclusion, the device provides continuous

single-mode coverage of nearly the full C+L span via controlled Vernier selection, rather than continuous tuning of one longitudinal mode.

The higher output near 1608 nm, despite the Er emission peak around 1532 nm (Fig. 2B), follows from the wavelength dependence of the net small-signal gain ($g_0(\lambda) = \Gamma[N_2\sigma_e(\lambda) - N_1\sigma_a(\lambda)] - \alpha(\lambda)$): Passive loss (dominated by Rayleigh scattering, $\propto \lambda^{-4}$) and reabsorption both decrease with wavelength, shifting the net gain to the L band. In our present EDWL devices (total Er concentration $5.6 \times 10^{26} \text{ m}^{-3}$ and 17 cm gain length, longer than optimal), incomplete inversion leads to additional reabsorption near 1530 nm that further suppresses output power. In addition, the Vernier filter FSR (~ 10 THz) is smaller than the Er emission bandwidth (~ 12 THz), so resonances near ~ 1530 nm and ~ 1610 nm can coincide within the Vernier envelope (Fig. 2C); combined with the L-band gain tilt, L-band modes reach threshold with a larger margin and dominate the output. The full C+L-band lasing in Fig. 3D is obtained by biasing the WDM loop mirror to attenuate L-band reflection, which raises L-band cavity loss and allows C-band modes to dominate. However, this bias simultaneously increases C-band loss, reducing the absolute C-band power. By implementing a Vernier filter with larger FSR and higher transmission, lowering C-band propagation loss, minimizing excited-state absorption, and reducing the effective Er-pair fraction (e.g., lower implantation dose with a longer gain waveguide), the EDWL should achieve more uniform power across the full C+L band.

In Q11, we present a complementary analysis of the EDWL output power dependence on key parameters, showing reduced power near 1530 nm and higher, comparable output from 1560 nm to 1610 nm, consistent with the above discussion. Recent measurements further confirm high-power lasing up to 47.6 mW at 1561 nm, supporting the interpretation that reabsorption decreases toward longer wavelengths.

[R4] G. Lihachev, et al. Frequency agile photonic integrated external cavity laser. *APL Photonics* 9.12 (2024).

Action taken:

- We clarify the interpretation of “mode-hop-free” behavior in the revised manuscript:

“The discrete lasing in Fig. 3D corresponds to the Vernier microresonator FSR. Although the laser is not strictly mode-hop-free (i.e., it does not continuously tune a single longitudinal mode), simultaneous tuning of both rings enables continuous [39] and deterministic single-mode coverage across the C + L band through Vernier selection, without uncontrolled mode hopping.”

[39] G. Lihachev, et al. Frequency agile photonic integrated external cavity laser. *APL Photonics* 9.12 (2024).

- We explain why L-band output exceeds C-band, analyze the causes, and propose strategies to raise C-band power:

“Nevertheless, a decrease in C-band power relative to the L-band is observed despite stronger Er emission near 1530 nm. This is primarily due to the limited Vernier FSR (≈ 80 nm; Fig. 2C), which co-aligns C- and L-band resonances (e.g., 1530/1610 nm) and biases lasing toward the L-band where passive loss and absorption are lower, yielding higher net gain $g - \alpha$. The near-full C+L lasing in Fig. 3D is obtained by bidirectional pumping to raise inversion and by biasing the WDM loop mirror to suppress L-band modes; however, this also increases the loss seen by co-aligned C-band modes, limiting the output power.

A wavelength-resolved model in Supplementary Note 5-6, including $\alpha(\lambda)$, Vernier transmission, ion-pair quenching, quantifies the measured power trend with wavelength. In the ideal limit (no quenching, wavelength-independent loss, unity Vernier transmission), the EDWL demonstrates near-uniform spectral behavior. Therefore, to enhance C-band output, we target reduced short-wavelength loss, a larger Vernier FSR and transmission, and a lower effective pair fraction (e.g., reduced Er dose with longer gain length). ”

- Figure 3C has been updated to show high-power lasing (47.6 mW) at 1561.82 nm, demonstrating strong device performance beyond 1608 nm:

Figure 3: Updated Fig. 3C showing 47.6 mW output power at 1561.82 nm.

11) The slope efficiency is 24% and the author suggest reduction in cavity and propagation loss to improve it, which is true but the loss seems already very low, and the figure in supplementary S6a suggest power increase of only 10 mW for loss decrease from 10 dB/m to 1 dB/m for 10 dB gain (which get's worst for lower gain), so it seems it's not possible to increase the power substantially.

The output power simulation in Fig. S6a varied only the propagation loss while keeping all other cavity terms fixed (Vernier insertion, mirror reflectivities). Under such constraint, reducing propagation loss from 10 to 1 dB/m yields only a modest ΔP_{out} because the round-trip loss is then dominated by fixed elements and the Er medium saturates at finite P_{sat} . Thus Fig. S6a should not be interpreted as a limit on achievable power.

To identify the leading contributors to the output power, we built a travelling-wave, self-consistent Er-rate-equation ($\text{Er}^{3+} 4I_{13/2} \leftrightarrow 4I_{15/2}$) solver that propagates forward and backward pump and signal:

$$\frac{dN_2}{dt} = -\frac{N_2}{\tau} + (N_1\sigma_{s,12} - N_2\sigma_{s,21})\phi_s + (N_1\sigma_{p,12} - N_2\sigma_{p,21})\phi_p$$

$$\frac{dN_1}{dt} = \frac{N_2}{\tau} + (N_2\sigma_{s,21} - N_1\sigma_{s,12})\phi_s + (N_2\sigma_{p,21} - N_1\sigma_{p,12})\phi_p$$

where $\sigma_{s,21}$, $\sigma_{p,21}$ and $\sigma_{s,12}$, $\sigma_{p,12}$ are the emission and absorption cross sections (taken from [R3]), ϕ_s , ϕ_p the ion flux, with $\phi_s = \frac{I_s}{h\nu_s}$ and $\phi_p = \frac{I_p}{h\nu_p}$. I_s and I_p are ion-weighted intensities and are calculated as $I_s = \Gamma_s \frac{P_s}{A_{s,\text{eff}}}$ and $I_p = \Gamma_p \frac{P_p}{A_{p,\text{eff}}}$, considering that only the portion of the optical field overlapping with the doped region contributes to the stimulated processes. At steady state, $\frac{dN_2}{dt} = \frac{dN_1}{dt} = 0$, the population inversion at point z is:

$$n(z) = \frac{N_2}{N_1 + N_2} = \frac{\tau\sigma_{s,12} \frac{I_s}{h\nu_s} + \tau\sigma_{p,12} \frac{I_p}{h\nu_p}}{\tau(\sigma_{s,12} + \sigma_{s,21}) \frac{I_s}{h\nu_s} + \tau(\sigma_{p,12} + \sigma_{p,21}) \frac{I_p}{h\nu_p} + 1}$$

For a given pump, forward/backward pump powers are propagated with passive loss to obtain the pump-only inversion $n(z)$ for threshold evaluation: The round-trip small-signal factor $G_0 = R_1 R_2 T_f^2 \exp[2 \int (g_{\text{eff}}(z) - \alpha) dz]$ determines threshold (from Eq. 4 in Supplementary Information), where $g_{\text{eff}}(z) = \Gamma \left(n(z)(1 - n_c)\sigma_{21} - \left((1 - n(z))(1 - n_c) + n_c \right) \sigma_{12} \right) N_0$ is the effective gain at position z , and $n_c = 1 - \frac{N_1 + N_2}{N_0}$ denotes the fraction of non-emitting erbium ions arising from pair-induced quenching at high Er concentrations, a mechanism likely present in our lasers. Above threshold ($G_0 > 1$), forward/backward signal and pump are propagated as:

$$\frac{dI(z)}{dz} = g(z)I(z) - \alpha I(z)$$

with boundary conditions $P_+(0) = R_1 P_-(0)$ and $P_-(L) = R_2 P_+(L)$. The inversion is updated from intensities $I_s(z) = \Gamma_s [P_+(z) + P_-(z)] / A_{s,\text{eff}}$ and $I_p(z) = \Gamma_p [P_{p,\text{fwd}}(z) + P_{p,\text{bwd}}(z)] / A_{p,\text{eff}}$ until convergence, while the left boundary is enforced by bisection on $P_+(0)$. In the fabricated device, mirror 1 is WDM-like, whereas in the simulations we deliberately approximate mirror 1 as a broadband reflector identical to mirror 2, with 80% pump reflectivity, to model an effectively single-sided-pumped cavity with enhanced pump recycling. The Vernier filter transmission is applied once per pass. The right-port output is $P_{\text{out}} = (1 - R_2) P_+(L)$.

Here we present the dependence of P_{out} on passive propagation loss α , loop-mirror reflectivities R_1 and R_2 , Vernier-filter transmission T_f , Er-ion clustering ratio n_c , Er concentration N_0 , and gain length, at a fixed on-chip pump power of 200 mW:

Figure R9: Parametric sensitivity of the output power P_{out} at 1550 nm, at a fixed on-chip pump power of 200 mW. (a) Output power as a function of passive loss α (dB/m) with Vernier filter transmission $T_f = 0.9$, $R_1 = 1$, $R_2 = 0.2$, non-emitting (parasitic) fraction $n_c = 0.1$. The pump loss is fixed to 1 dB/m. (b) Output power as a function of Vernier filter transmission T_f , with $\alpha = 1$ dB/m, $R_1 = 1$, $R_2 = 0.2$, $n_c = 0.1$. (c) Output power as a function of R_1 , with $\alpha = 1$ dB/m, $T_f = 0.9$, $R_2 = 0.2$, $n_c = 0.1$. (d) Output power as a function of R_2 , with $\alpha = 1$ dB/m, $T_f = 0.9$, $R_1 = 1$, $n_c = 0.1$. (e) Output power as a function of non-emitting ion fraction n_c , with $\alpha = 1$ dB/m, $T_f = 0.9$, $R_1 = 1$, $R_2 = 0.2$. (f) Output power as a function of Er concentration N_0 , with $\alpha = 1$ dB/m, $T_f = 0.9$, $R_1 = 1$, $R_2 = 0.2$, for $n_c = 0, 0.05$ and 0.1 . (a-f) assume the implanted Er concentration $N_0 = 5.6 \times 10^{26} \text{ m}^{-3}$. (g) Simulated wavelength-dependent output power, based on measured passive loss and Vernier transmission from fabricated devices (blue trace, loop mirrors: $R_1 = 1, R_2 = 0.2, n_c = 0.1, N_0 = 5.6 \times 10^{26} \text{ m}^{-3}$), comparing with a constant propagation loss of 1 dB/m (red trace). This simulation also considers SiO_2 cladding with H-related absorption near 200 THz (yellow trace, with pump loss of 20 dB/m), and an ideal case (green trace, $\alpha = 1$ dB/m, $T_f = 1$, $R_1 = 1, R_2 = 0.2, n_c = 0, N_0 = 5.6 \times 10^{26} \text{ m}^{-3}$). (h) Output power as a function of gain spiral length, based on measured passive loss and Vernier transmission from fabricated devices, showing an optimum around 0.08-0.1 m at an Er concentration of $N_0 = 5.6 \times 10^{26} \text{ m}^{-3}$, with n_c varying from 0 to 0.1.

Simulations in Fig. R9 show that output power is governed by several cavity parameters, not propagation loss alone: P_{out} rises with lower propagation loss α and higher Vernier transmission T_f . Maximizing the reflection of mirror 1 ($R_1 \rightarrow 1$) monotonically increases power by reducing round-trip loss, while the output mirror R_2 exhibits a low-reflectivity optimum, where relatively weak output coupling maximizes power by balancing extraction with intracavity build-up. We observe that even a small non-emitting fraction n_c strongly suppresses power, motivating mitigation of Er clustering. At fixed 17 cm gain length, lowering the Er concentration raises output by reducing clustering (Fig. R9(f)); at a fixed Er concentration of $N_0 = 5.6 \times 10^{26} \text{ m}^{-3}$, the optimal gain length is ~ 8 cm (Fig. R9(h)). Including the measured wavelength dependences of α , T_f , and Er cross sections ($\sigma_{s,12}$ and $\sigma_{s,21}$), the model reproduces the experimentally observed lower C-band power relative to the L-band. In an idealized device ($n_c = 0$, $\alpha = 1 \text{ dB/m}$, $T_f = 1$), the predicted P_{out} is nearly flat over the Er emission band (Fig. R9(g)).

The slope efficiency η_s is obtained by linear fit of P_{out} versus pump power just above threshold:

Figure R10: **Parametric sensitivity of the slope efficiency η_s at 1550 nm.** (a) Slope efficiency as a function of passive loss α (dB/m) with Vernier filter transmission $T_f = 0.9$, $R_1 = 1$, $R_2 = 0.2$, non-emitting (parasitic) fraction $n_c = 0.1$. The pump loss is fixed to 1 dB/m. (b) Slope efficiency as a function of Vernier transmission T_f , $\alpha = 1 \text{ dB/m}$, $R_1 = 1$, $R_2 = 0.2$, $n_c = 0.1$. (c) Slope efficiency as a function of R_1 , with $\alpha = 1 \text{ dB/m}$, $T_f = 0.9$, $R_2 = 0.2$, $n_c = 0.1$. (d) Slope efficiency as a function of R_2 , with $\alpha = 1 \text{ dB/m}$, $T_f = 0.9$, $R_1 = 1$, $n_c = 0.1$. (e) Slope efficiency as a function of non-emitting ion fraction n_c , with $\alpha = 1 \text{ dB/m}$, $T_f = 0.9$, $R_1 = 1$, $R_2 = 0.2$. (f) Slope efficiency as a function of Er concentration N_0 , with $\alpha = 1 \text{ dB/m}$, $T_f = 0.9$, $R_1 = 1$, $R_2 = 0.2$, for $n_c = 0, 0.05$ and 0.1. (a-f) assume the implanted Er concentration $N_0 = 5.6 \times 10^{26} \text{ m}^{-3}$. (g) Simulated output-pump characteristic for extraction of on-chip slope efficiency $\eta_s = dP_{\text{out}}/dP_{\text{pump}}$ at 1550 nm, using measured wavelength-dependent passive loss and Vernier transmission extracted from the device (loop mirrors $R_1 = 1$, $R_2 = 0.2$, $n_c = 0.1$, $N_0 = 5.6 \times 10^{26} \text{ m}^{-3}$) and ideal values in simulation ($\alpha = 1 \text{ dB/m}$, $T_f = 1$, $R_1 = 1$, $R_2 = 0.2$, $n_c = 0$, $N_0 = 5.6 \times 10^{26} \text{ m}^{-3}$). The slope efficiency is obtained from a linear fit to data points above threshold. (h) Simulated wavelength dependent slope efficiency using measured loss and Vernier filter transmission. Fluctuations arise from variation from fitted external coupling rate in test resonators.

The dominant levers for higher slope efficiency η_s are minimizing Vernier insertion loss, using high R_1 and low R_2 , suppressing clustering (small n_c), and selecting an optimal N_0 , similar to the ones of P_{out} . Figure R10(h) shows a 1520-1630 nm slope efficiency sweep, revealing that slope efficiency peaks at wavelengths longer than the Er emission maximum (~ 1535 nm): Near the C-band the required transparency inversion is

high because both absorption and emission cross sections are large; much of the pump is therefore spent merely overcoming reabsorption (including from the non-emitting pool), which depresses η_s . Toward longer wavelengths $\sigma_{12}(\lambda)$ falls faster than $\sigma_{21}(\lambda)$ so reabsorption weakens and the incremental signal gain per pump photon increases. At the same time the passive waveguide loss $\alpha(\lambda)$ drops with longer wavelength, improving the internal efficiency factor. The Vernier filter also helps, at zero detuning, the filter transmission is (Supplementary Section 6):

$$T_f = \left| \frac{\kappa_{ex,1} \cdot \kappa_{ex,2}}{\left(\frac{\kappa_0}{2} + \kappa_{ex,1}\right) \cdot \left(\frac{\kappa_0}{2} + \kappa_{ex,2}\right)} \right|^2$$

so any reduction of $\kappa_0(\lambda)$ with wavelength increases transmission and raises the external slope. Consequently, the simulation predicts a broad maximum in η_s around ~ 1570 - 1600 nm where reabsorption is lower, passive/filter losses are smaller, and pump power is converted to output most efficiently.

We measured the EDWL slope efficiency at various wavelengths (Fig. R11) to validate the simulation in Fig. R10(h), observing the same trend of reduced efficiency and higher threshold pump power near 1550 nm.

Figure R11: Measured slope efficiency of the EDWL at different wavelengths.

To further increase slope efficiency, a spectrally narrow 1480 nm pump should be used to maximize overlap with the $\text{Er}^{3+} {}^4\text{I}_{15/2} \rightarrow {}^4\text{I}_{13/2}$ absorption and minimize off-resonant pump power [R5]. Our 1480 nm pump has a 3 dB spectral width of ~ 1 nm (Fig. R12), frequency components detuned from the peak are absorbed with lower cross-section. A narrower-band 1480 nm pump would increase the effective absorption coefficient and inversion in our EDWL, and thus can further improve the EDWL slope efficiency under otherwise identical power conditions.

Alternatively, a 980 nm pump can be used to further improve the efficiency due to its higher absorption cross section and reduced excited-state absorption in erbium, leading to more efficient population inversion. Our current 1480 nm pumping scheme is a compromise due to the edge-coupler design and propagation loss.

Figure R12: Optical spectrum of the 1480 nm pump laser (QPhotonics QFBGLD-1480-500).

[R3] Y. Liu, et al. A photonic integrated circuit–based erbium-doped amplifier. *Science* 376.6599: 1309-1313 (2022).

[R5] E. Desurvire, et al. Efficient erbium-doped fiber amplifier at a 1.53- μm wavelength with a high output saturation power. *Optics Letters* 14.22: 1266-1268 (1989).

Action taken:

- We add the Vernier filter transmission analysis to the Supplementary Information:

“We present the simulation and measurement of the Vernier-filter transmission.

The Vernier filter transmission, $T_f = |s_{\text{out}}|^2$, is given by (see Eq. 12, derived in Section 7):

$$T_f = \left| \frac{\kappa_{\text{ex},1} \kappa_{\text{ex},2}}{(i\Delta_1 + \kappa_1) \cdot (i\Delta_2 + \kappa_2)} \right|^2, \quad \kappa_j = \frac{\kappa_0}{2} + \kappa_{\text{ex},j} + \kappa_{\text{p},j} \quad (j = 1, 2)$$

where Δ_j is the detuning of resonator j , $\kappa_{\text{ex},j}(\lambda)$ are the external coupling rates, $\kappa_0(\lambda)$ is the intrinsic loss rate, and $\kappa_{\text{p},j}$ account for parasitic coupling (higher-order and radiation modes).

In Fig. S6(a), we measure the wavelength-dependent κ_{ex} using a test resonator that shares the Vernier geometry but employs a single bus coupling with a 2 μm gap. Vernier resonator κ_{ex} values are then calibrated from the simulated κ_{ex} -gap relation (Lumerical FDTD). Figure S6(b) shows the simulated Vernier transmission versus the normalized coupling ($\kappa_{\text{ex},j}/\kappa_0$): increasing $(\kappa_{\text{ex},j}/\kappa_0)$ for both resonators raises the peak filter transmission. Using the calibrated κ_{ex} and κ_0 from Fig. 2D, Fig. S6(c) shows Vernier filter transmission at different wavelengths, yielding 0.6-0.7 in the C and L bands. Higher transmission is attainable with larger κ_{ex} (smaller gaps) or reduced κ_0 .

Figure S6: Vernier filter loss analysis. (a) Measured external coupling rate $\kappa_{\text{ex}}/2\pi$ versus wavelength for a test ring resonator sharing the Vernier ring geometry but using a single bus waveguide and a 2 μm gap. Vernier ring κ_{ex} values are calibrated via a κ_{ex} -gap

relation simulated using Lumerical FDTD. (b) Simulated Vernier filter transmission as a function of κ_{ex}/κ_0 for both resonators, evaluated at zero detuning. (c) Fitted transmission of Vernier filter in the EDWL.

- We update the Supplementary section 4 with new simulation results:

“To capture spatial effects such as gain depletion and power variation along the erbium-doped waveguide, the model is extended to a distributed form, $A(z,t)$, implemented through a travelling-wave, self-consistent Er^{3+} -rate-equation (${}^4I_{13/2} \leftrightarrow {}^4I_{15/2}$) solver that propagates the forward and backward pump and signal...”

- We update the Supplementary section 4 with new simulation results on output power, slope efficiency, and Schawlow-Townes limited laser linewidth:

“Using Eq.6 and Eq.8, we simulate the Schawlow-Townes laser linewidth: $\Delta\nu_{ST} = n_{sp} \frac{h\nu}{4\pi P_{out}} (\Delta\nu_c)^2$, where $\Delta\nu_c = \frac{1}{2\pi\tau_{ph}}$ is the cavity loss rate, with τ_{ph} the photon lifetime. ...”

Figure S12: **Parametric sensitivity of Schawlow-Townes linewidth.** (a) Schawlow-Townes linewidth as a function of passive loss α (dB/m) with Vernier filter transmission $T_f = 0.9$, $R_1 = 1$, $R_2 = 0.2$, non-emitting (parasitic) fraction $n_c = 0.1$. The pump loss is fixed to 1 dB/m. (b) Schawlow-Townes linewidth as a function of Vernier filter transmission T_f , $\alpha = 1$ dB/m, $R_1 = 1$, $R_2 = 0.2$, $n_c = 0.1$. (c) Schawlow-Townes linewidth as a function of R_1 , with $\alpha = 1$ dB/m, $T_f = 0.9$, $R_2 = 0.2$, $n_c = 0.1$. (d) Schawlow-Townes linewidth as a function of R_2 , with $\alpha = 1$ dB/m, $T_f = 0.9$, $R_1 = 1$, $n_c = 0.1$. (e) Schawlow-Townes linewidth as a function of non-emitting ion fraction n_c , with $\alpha = 1$ dB/m, $T_f = 0.9$, $R_1 = 1$, $R_2 = 0.2$. (f) Schawlow-Townes linewidth as a function of Er concentration N_0 , with $\alpha = 1$ dB/m, $T_f = 0.9$, $R_1 = 1$, $R_2 = 0.2$, for $n_c = 0, 0.05$ and 0.1 . (a–f) assume the implanted Er concentration $N_0 = 5.6 \times 10^{26} \text{ m}^{-3}$. (g) Simulated wavelength-dependent Schawlow-Townes linewidth, based on measured passive loss and Vernier transmission from fabricated devices (blue trace, loop mirrors: $R_1 = 1$, $R_2 = 0.2$, $n_c = 0.1$, $N_0 = 5.6 \times 10^{26} \text{ m}^{-3}$), comparing with a constant propagation loss of 1 dB/m and a Vernier filter transmission of 0.9 (red trace). (h) Schawlow-Townes linewidth as a function of gain spiral length, based on measured passive loss and Vernier transmission from fabricated devices, showing an optimum around 0.08–0.1 m at an Er concentration of $N_0 = 5.6 \times 10^{26} \text{ m}^{-3}$, with n_c varying from 0 to 0.1.

12) Fig. 4 has interesting data but it seems not clear what the actual linewidth degradation is, please show it in the freq. spectrum (2 d plot – freq. vs power) at different reflection points (at least show at some important points – low reflection, mid-level reflection, high level reflection).

The linewidth measurement results are summarized in Figure R13. Under external injection, the DSH spectrogram remains essentially unchanged across the tested input powers (Fig. R13(a)) within a detuning under 217 MHz (Fig. R13(d)), where the derived frequency-noise spectra $S_{\delta\nu}$ show no systematic linewidth degradation (Fig. R13(g)).

With increasing self-reflection, the low-offset frequency noise decreases progressively (Fig. R13(h); 35, 5, and 0 dB attenuation), and the integrated linewidth narrows monotonically, approaching that of a tabletop ECDL at the highest return. At high self-reflection (<15 dB attenuation), we observe that the EDWL becomes phase-sensitive and intermittently switches between a noise-broadened state and a coherent, narrow-line state, visible in the spectrograms (Fig. R13(b)) and beat notes (Fig. R13(e)(f)). This bistability is consistent with delayed self-feedback in a compound cavity (Lang-Kobayashi regime): slow drift of the external-path phase or polarization changes the effective feedback phase and magnitude, shifting the laser between coexisting stable external-cavity modes and producing mode hops or hysteresis [R6-7]. The state switching can be eliminated or strongly mitigated by stabilizing the feedback phase and polarization (shorter, temperature-stabilized PM loop and mechanical isolation) [R7].

Because the broadened state is non-stationary and non-Lorentzian, a linewidth is not well defined; therefore linewidths in Fig. R13(h) are reported for the coherent state only. The observed linewidth narrowing with increasing self-injection is expected for our Er-doped Si_3N_4 laser with a small amplitude-phase coupling (small Henry factor α_H). The reinjected, time-delayed field provides passive negative feedback at low offset frequencies (delay limited with corner frequency $f_c \sim \frac{1}{2\pi\tau}$ [R8-9]) with a closed-loop suppression $\propto |1 + Ge^{-j2\pi f\tau}|^{-2}$, where G scales with the returned field amplitude [R6-7]. In addition, the returned field lowers the effective mirror loss and increases the photon lifetime τ_{ph} , reducing the Schawlow-Townes contribution.

[R6] R. Lang and K. Kobayashi. External optical feedback effects on semiconductor injection laser properties. *IEEE journal of Quantum Electronics* 16.3: 347-355 (1980).

[R7] J. Mork, et al. Bistability and low-frequency fluctuations in semiconductor lasers with optical feedback: a theoretical analysis. *IEEE journal of Quantum Electronics* 24.2, 123-133 (2002).

[R8] M. Lipka, et al. Optical frequency locked loop for long-term stabilization of broad-line DFB laser frequency difference. *Applied Physics B* 123.9: 238 (2017).

[R9] Y. Chao et al. Robust suppression of high-frequency laser phase noise by adaptive Pound-Drever-Hall feedforward. *Physical Review Applied* 23.1: L011005 (2025).

Action taken:

- We add this measurement as a new section, “**EDWL frequency noise response to external injection and self-reflection**”, in the Supplementary information:

“In this section, we present EDWL frequency-noise measurements as a function of the reflection level for external injection and for self-reflection (Figs. S17(a)-(c)).

With external injection detuned by less than 217 MHz (Fig. S17(d)), the DSH spectrogram remains unchanged across the tested powers up to 10 dBm (Fig. S17(a)), and the corresponding frequency-noise spectra show no systematic linewidth degradation (Fig. S17(g)). This stability can be explained by the Vernier filter’s microresonator drop ports, which function as intrinsic bandpass filters and inherently suppress back-reflections.

In contrast, increasing self-reflection produces a progressive reduction of low-offset frequency noise; the integrated linewidth decreases monotonically and, at the highest return (full self-reflection), approaches that of a tabletop ECDL used as the local oscillator (Fig. S17(h)). We observe that, at strong self-reflection (<15 dB attenuation), the EDWL intermittently switches between a noise-broadened state and a narrow-line coherent state, which are visible in the DSH spectrum (Fig. S17(b)) and beat-note traces (Fig. S17(e)(f)). This bistability is consistent with delayed self-injection in a compound cavity described by the Lang-Kobayashi model: slow drifts of feedback phase (and polarization-dependent return) move the operating point between stable external-cavity solutions and neighboring external-cavity modes [S16][S17]. Because the broadened state is non-stationary and non-Lorentzian, a linewidth is not well defined; we therefore report linewidths for the coherent state.

Figure R13: Frequency noise response of the EDWL to external injection and self-reflection. (a) Spectrogram of the delayed self-heterodyne (DSH) beatnote while increasing the injected external-laser power. (b) Spectrogram of the DSH beatnote while increasing the self-reflection (return via circulator and VOA; higher return at lower attenuation). (c) Experimental setup for frequency noise measurement with external and internal injection. (d) Optical heterodyne beatnote between the EDWL and the injected laser, showing

a detuning $\Delta\nu=216.53$ MHz. (e) Heterodyne beatnote between the EDWL and a Toptica ECDL used as the local oscillator (LO) at 0 dB self-reflection attenuation, showing a noise-broadened (perturbed) state. (f) A coherent (narrow-line) state at 0 dB self-reflection attenuation. (g) EDWL frequency noise spectra for two external-injection points marked in (a): state 1 (0 mW) and state 2 (9.8 mW). (h) EDWL frequency noise for three self-reflection levels marked in (b): state 3 (35 dB), state 4 (5 dB), and state 5 (0 dB), showing progressive low-offset noise reduction with increasing return. The LO's own frequency-noise spectrum (Toptica ECDL, measured against a Koheras fiber laser) is shown in grey for reference. At the coherent state with full self-reflection, the EDWL frequency noise approaches that of the tabletop ECDL.

The observed linewidth narrowing with increasing self-reflection follows the expected action of coherent optical negative feedback. The reinjected field is a time-delayed copy of the intracavity field; for offset frequencies below the delay corner $f_c \sim \frac{1}{2\pi\tau}$ [S18][S19], the feedback acts as passive phase regulation, yielding a closed-loop suppression factor that, in linearized form, scales as $|1 + Ge^{-j2\pi f\tau}|^{-2}$ [S16][S17]. This mechanism and the associated linewidth-narrowing relations under weak feedback are well established in the cited literature.

In summary, the linewidth reduction in our EDWL at increasing self-reflection is different from ECDLs, which have a large linewidth-enhancement factor α_H and are very sensitive to uncontrolled optical feedback. In ECDLs with added reflection, the intensity and phase coupling often drive coherence collapse. Thus, ECDLs use optical isolation to suppress feedback and stabilize single-frequency operation. In our EDWL, the frequency noise reduce monotonically with increased self-reflection. Part of the observed frequency-noise reduction arises from an increased effective photon lifetime due to the external reflection (extended effective cavity), together with delayed self-injection that provides negative feedback on phase fluctuations and suppresses low-frequency frequency noise.”

[S16] R. Lang and K. Kobayashi. External optical feedback effects on semiconductor injection laser properties. *IEEE journal of Quantum Electronics* 16.3: 347-355 (1980).

[S17] J. Mork, et al. Bistability and low-frequency fluctuations in semiconductor lasers with optical feedback: a theoretical analysis. *IEEE journal of quantum electronics* 24.2, 123-133 (2002).

[S18] M. Lipka, et al. Optical frequency locked loop for long-term stabilization of broad-line DFB laser frequency difference. *Applied Physics B* 123.9: 238 (2017).

[S19] Y. Chao et al. Robust suppression of high-frequency laser phase noise by adaptive Pound-Drever-Hall feedforward. *Physical Review Applied* 23.1: L011005 (2025).

- We refer to this SI section in the main text:

“The detailed frequency noise characterization of the EDWL under external injection and self-reflection is presented in Supplementary Note 10. The measurements show that external injection, detuned by 217 MHz, does not measurably affect the DSH spectrogram or linewidth, confirming the robustness of the Vernier-filter design against back-reflections. In contrast, increasing self-reflection reduces low-offset frequency noise and narrows the integrated linewidth, consistent with coherent negative feedback via delayed self-injection (Lang-Kobayashi model, [52-55]), particularly in devices with a small linewidth-enhancement factor α_H , such as our EDWL.”

13) Also do you expect nonlinear effect with this CW laser power like four wave mixing in the resonator? Which brings to another question what is quality factor of the resonators used?

In our present devices, including the highest-power laser (47.6 mW off-chip; updated Fig. 3C), we observe no clear signatures of four-wave mixing (FWM) under normal operating conditions. We further demonstrate high-power lasing without FWM in Fig. R14 (the observed spectral sidelobes originate from adjacent Vernier modes, not nonlinear mixing):

Figure R14: Examples of high-power lasing with side-mode suppression ratio (SMSR) exceeding 79 dB. No evidence of four-wave mixing (FWM) is observed; the residual side modes originate from neighboring Vernier resonator modes.

While Si_3N_4 resonators with high Q can, in principle, support FWM due to strong intracavity field buildup, the effect is strongly suppressed in our devices by the large mode area A_{eff} and the strong normal group-velocity dispersion ($\beta_2 > 0$) of the 200 nm Si_3N_4 waveguide near $1.55 \mu\text{m}$. For degenerate Kerr FWM with a CW pump at ω_p , signal/idler appear at $\omega_{s,i} = \omega_p \pm \Omega$. Efficiency is set by the phase mismatch:

$$\Delta k(\Omega) \approx -\beta_2\Omega^2 - 12\beta_4\Omega^4 + \dots + 2\gamma P_{\text{eff}}$$

Here, the Kerr-induced shift $2\gamma P_{\text{eff}}$ is too small to compensate the strong normal dispersion, preventing phase matching and thus suppressing parametric gain. Together with the relatively low nonlinear coefficient ($\gamma = n_2\omega/(cA_{\text{eff}})$), Fig. R15(b) in our weakly confined waveguide design, these factors preclude efficient four-wave-mixing in our laser cavities.

Figure R15: (a) Fitted GVD of a 200 nm thick Si_3N_4 microresonator. (b) Simulation of nonlinear coefficient γ of 200 nm thick Si_3N_4 waveguides with varying waveguide widths.

Regarding resonator quality factors, we extract $Q_i = \frac{\omega}{\kappa_0} = \frac{n_g\omega}{\alpha c}$ from the measured propagation loss in Fig. 2D:

Figure R16: Measured quality factor of a 50 GHz FSR Si_3N_4 resonator.

This yields intrinsic quality factors $Q_i \approx (8-18) \times 10^6$. For the Vernier resonator, which is coupled to both through and drop bus waveguides with equal external rates $\kappa_{\text{ex}} \approx 15 \kappa_0$, the loaded quality factor can be estimated as:

$$Q_L = \frac{Q_i}{1 + 2 (\kappa_{\text{ex}}/\kappa_0)} \approx \frac{Q_i}{31}.$$

Action taken:

We have added a note on the possibility of FWM to the manuscript:

“We observe no four-wave mixing (FWM) sidebands in current devices, including at the highest output power of 47.6 mW (Fig. 3C). Although Si₃N₄ resonators with high Q can in principle support FWM due to strong intracavity field buildup, the strong normal group-velocity dispersion and large mode area of the 200 nm waveguides prevent efficient phase matching and thus suppress nonlinear interactions under our operating conditions.”

14) Also the concentration of $5.6 \times 10^{26} \text{ m}^{-3}$ seems a bit high, do you expect quenching effects to kick in which rare-earth ions are known for? It seems that this is not a problem as in the supplementary authors simulate with $9.4 \times 10^{26} \text{ m}^{-3}$, please present experimental data if possible on performance with different level concentration.

Our direct ion implantation yields a Gaussian depth profile with excellent lateral uniformity, suppressing local clustering relative to co-deposition or diffusion methods. Consistent with limited quenching, we measure lifetimes of 3.4 ms at Er concentration of $5.6 \times 10^{26} \text{ m}^{-3}$ and 2.7 ms at $9.4 \times 10^{26} \text{ m}^{-3}$ in Er:Si₃N₄ waveguides. The EDWL performance is dominated by pump depletion and parasitic absorption rather than lifetime shortening.

To quantify clustering effects, in our reply to Q11 we model the concentration-induced quenching as a parasitic fraction n_c of Er ions that absorb but do not emit, thereby increasing loss without contributing gain. Simulations in Fig. R9(e)(f) show a monotonic power penalty: increasing n_c from 0 to 0.3 reduces on-chip output power by ~15 dB. Our process-relevant $n_c \lesssim 0.05-0.1$, fitted from slope efficiency in Fig. R10(g), implies only a modest penalty at current concentrations.

Experimentally, an EDWL with $9.4 \times 10^{26} \text{ m}^{-3}$ exhibits lasing power and tuning range comparable to the $5.6 \times 10^{26} \text{ m}^{-3}$ device (Fig. R3) under similar on-chip pump (~180 mW, unidirectional), with reduced C-band output consistent with the simulated wavelength-dependent impact of pair-induced parasitic absorption (Fig. R17). A systematic study of Er clustering and quenching versus concentration will be reported separately.

Figure R17: Lasing map of a high-Er concentration EDWL ($9.4 \times 10^{26} \text{ m}^{-3}$), showing tunability and reduced C-band power compared to the $5.6 \times 10^{26} \text{ m}^{-3}$ device.

Action taken:

We include the high-concentration EDWL lasing measurement and a discussion of concentration-dependent quenching in the Supplementary Information:

“Our direct ion implantation yields a Gaussian depth profile with excellent lateral uniformity, suppressing local clustering relative to co-deposition or diffusion methods. Consistent with limited quenching, we measure lifetimes of 3.4 ms at Er peak concentration of $5.6 \times 10^{26} \text{ m}^{-3}$ and 2.7 ms at $9.4 \times 10^{26} \text{ m}^{-3}$ in Er:Si₃N₄ waveguides. The EDWL performance is dominated by pump depletion and parasitic absorption rather than lifetime shortening.

We characterized an EDWL with a higher total concentration at $9.4 \times 10^{26} \text{ m}^{-3}$. With ~180 mW unidirectional on-chip pump at 1480 nm, it lases across the C+L bands with output power and tuning range comparable to those of devices with $5.6 \times 10^{26} \text{ m}^{-3}$ Er concentration (Fig. S23). A notable difference is a reduced C-band output relative to the lower-concentration device, which is consistent with our simulations of wavelength dependence in the presence of Er ion clustering (Fig. S8(e)(f)), indicating a stronger concentration-dependent quenching at higher Er doping.”

15) In the supplementary eq.4 is solved in time domain or should it be calculated time independently in the steady state (as the laser is in the steady state), and shouldn't the calculation be as a function of length as shown in the supplementary Fig.S5b. Also are the eq.8 – 18 in the supplementary never explored before (please give reference to these equations).

Equation (4) in the Supplementary Information provides a lumped mean-field description, where the distributed gain, loss, and phase evolution along the cavity are averaged over one round trip into a single global balance. In steady state ($dA/dt = 0$), the condition $g = \alpha + \frac{2}{L} \ln \frac{1}{\sqrt{R_1} \sqrt{R_2} T_f}$ defines the lasing threshold and above-threshold gain clamping. The amplitude $A(t)$ thus represents the round-trip-averaged intracavity field envelope.

Since $P = |A|^2$, $\frac{dP}{dt} = 2 \text{Re} \left(A^* \cdot \frac{dA}{dt} \right)$, Eq. (4) is equivalently:

$$\frac{n_g L}{c} \frac{1}{P} \frac{dP}{dt} = (gL - \alpha L) - \left[\ln \frac{1}{R_1} + \ln \frac{1}{R_2} + \ln \frac{1}{T_f^2} \right]$$

For cold cavity ($g = 0$), this reduces to $\frac{dP}{dt} = -\kappa P$, with $\kappa = \frac{c}{n_g L} \left(\alpha L + \ln \frac{1}{R_1 R_2 T_f^2} \right)$. Hence the time-domain mean-field equation connects directly to the cavity loss rate that enters the Schawlow-Townes linewidth $\Delta\nu \propto \kappa^2$.

In parallel, power growth/decay and gain clamping are inherently position-dependent in this cavity because the laser combines distributed processes (Er gain $g(z)$, propagation loss α with Vernier insertion, output coupling R_2) and counter-propagating modes. For quantitative power optimization, we solve the traveling-wave equations using a reduced steady-state model with saturable gain to obtain the length-dependent intracavity powers:

$$\frac{dP_{\pm}}{dz} = \pm(g - \alpha)P_{\pm}$$

with enforced boundary conditions $R_1 \cdot P_-(0) = P_+(0)$, $R_2 \cdot P_+(L) = P_-(L)$, and a single-pass filter transmission T_f . This z-resolved model captures the spatially varying power due to distributed gain/loss, filter insertion, output coupling and local gain $g(z)$. Integrating over one round trip yields:

$$1 = R_1 R_2 T_f^2 e^{[2 \int_0^L (g - \alpha) dz]}$$

which is the steady-state balance in Eq. (4), thus the two formulations are mathematically consistent.

The intracavity power distribution and wavelength dependent output power simulation in Fig S5 (b)-(d) are based on the z-resolved model. We keep Eq. (4) because it provides a global balance and dynamic interpretation, and connects directly to photon lifetime, while the z-resolved solution explains where power is gained, lost and extracted inside the cavity and gives directly P_{out} . Together they yield a self-consistent estimate for coherence and power optimization.

In Fig. R18, we present an updated simulation of intracavity power distribution based on the model described in Q11.

Equations 8-10 in Supplementary section 6 are the standard Langevin equations from input-output theory and have been extensively treated in quantum optics/cavity QED [R10] and widely used integrated photonics [R11]. Equations 11-18 give the output complex field amplitude and phase derived from Eqs. 8-10, no new physics is introduced beyond this derivation.

[R10] H. Haus, et al. Waves and fields in optoelectronics (Ch.7: coupling of modes-resonators and couplers) (1984).

[R11] M. Gorodetsky, et al. Optical microsphere resonators: optimal coupling to high- Q whispering-gallery modes. *Journal of the Optical Society of America B* 16.1: 147-154 (1999).

Figure R18: **Simulation of intra-cavity power distribution.** (a) Schematic of the laser cavity with two mirrors, an Er-doped gain section, and a Vernier filter. The 1480 nm pump is injected unidirectionally at the gain-section end. The simulated output is taken at mirror 2. (b) Simulated intra-cavity power distribution for forward (P_+) and backward (P_-) propagating light. Losses due to mirror transmission and filtering are highlighted.

Action taken:

- We update the simulation of intra-cavity power distribution in Supplementary section 4:

“Above threshold ($G_0 > 1$), the forward and backward signal and pump are propagated as:

$$\frac{dI(z)}{dz} = g(z)I(z) - \alpha I(z)$$

... Figure S7 (b) shows the simulated intracavity power distribution, where mirror transmissions and filter losses are highlighted. The forward-propagating field initially decreases, reaches a minimum, and then rises toward the end of the Er-doped gain section, while a similar trend is observed for the backward-propagating field. This behavior arises from the pumping scheme in Fig. S7(a): In the backward pumped configuration, $g(z)$ is initially small or negative and increases along z ; thus, the forward wave $P_+(z)$ attenuates in the weakly inverted region before amplifying where $g(z) > 0$. Conversely, the backward wave $P_-(z)$ grows near the pumped end and decays as it propagates into regions of lower inversion, reaching its maximum where $g(z) = 0$.”

- We now cite the references for Langevin equations at the start of Supplementary Section 6:

“The Langevin equations for the system [S12][S13] can be written as follows”

[S12] H. Haus, et al. Waves and fields in optoelectronics (Ch.7: coupling of modes-resonators and couplers) (1984).

[S13] M. Gorodetsky, et al. Optical microsphere resonators: optimal coupling to high-Q whispering-gallery modes. Journal of the Optical Society of America B 16.1: 147-154 (1999).

Reviewer #2

The authors present on wafer-scale Er-implanted SiN waveguide lasers with high powers, stable operation and wide tuning range. The main advances are implementing the lasers on a low confinement SiN platform (200 nm) with typical implant beam energies (< 500 keV), enabling mass production on 8 and 12 inch wafers. Although one might argue that the main overall ideas of Er ion implantation in SiN and Vernier lasing have been shown, the wafer scale integration of Er-implanted SiN lasers is a big step and it is combined with significantly improved performance. The paper is very well organized and the results are impressive and of general interest to the photonics community, and I recommend it be published in Nature Communications largely as it is. I just have some relatively minor comments:

We thank the reviewer for the positive assessment and recommendation for publication.

- it would be helpful to add information on the SiN refractive index vs. Er doping concentration. The changes should be small but can impact the device design and possibly reflections in the laser cavity, and it is also important for simulating devices on the platform.

We thank the reviewer for raising the question of the Si₃N₄ refractive index change with Er implantation. We provide spectroscopic ellipsometry measurements of the film index before and after implantation:

Figure R19: Refractive index modification in Si₃N₄ from Er implantation. (a) Measured refractive index of Si₃N₄ film before and after Er implantation. (b) Integrated dispersion $D_{int}/2\pi$ of a 22.5 GHz ring resonator before and after Er implantation.

The implantation details for the film under measurement are summarized in table R1:

Table R1: Er ion implantation parameters

Step 1	Step 3	Step 3	Total dose	Target peak concentration
350 keV	178 keV	81.2 keV	2.63×10^{15} ions/cm ²	3.25×10^{20} ions/cm ³
1.49×10^{15} ions/cm ²	7.09×10^{14} ions/cm ²	4.35×10^{14} ions/cm ²		

The atomic density of Si₃N₄ is:

$$N_{\text{Si}_3\text{N}_4} = \frac{\rho_{\text{Si}_3\text{N}_4} \cdot N_A}{M_{\text{Si}_3\text{N}_4}} \times 7 \approx 9.53 \times 10^{22} \text{ atoms/cm}^3$$

Atomic ratio of Er in the implanted film is therefore:

$$\frac{N_{\text{Er}}}{N_{\text{Si}_3\text{N}_4}} \approx 0.341\%$$

Figure R19(a) shows that implanting 0.341% Er slightly increases the Si₃N₄ film refractive index (from 1.9866 to 2.0106 at 1552 nm), indicating a modified material dispersion ($|dn/d\lambda|$).

Figure R19(b) quantifies the ring resonator waveguide mode dispersion $n_{\text{eff}}(\lambda)$ before and after implantation: the FSR ($D_1/2\pi$) increases from 22.544 GHz to 22.684 GHz, implying a $\sim 0.6\%$ decrease in group index n_g ($\text{FSR} = \frac{c}{n_g L}$). Note that the rings in the EDWL are undoped, the data shown here are from a doped test ring used to quantify implantation-induced refractive index changes.

The decrease in n_g does not contradict the increase in $n(\lambda)$: although n_{eff} increases slightly after implantation, the slope term in $n_g = n_{\text{eff}} - \lambda \frac{dn_{\text{eff}}}{d\lambda}$ near 1550 nm decreases enough to yield a net reduction in n_g . Small geometry/stress changes from implantation/anneal may also contribute to the waveguide-dispersion shift, but the overall modification is minor.

With higher Er concentration, the refractive index is expected to increase. Further investigation will be conducted in a separate study.

Action taken:

We added measured refractive indices of Si₃N₄ film before and after implantation in a new Supplementary section “**Refractive index modification of Si₃N₄ by Er implantation**”:

“To quantify the erbium-induced refractive-index change in Si₃N₄, we performed spectroscopic ellipsometry on blanket films before and after implantation. The implantation sequence consisted of three energy steps (350, 178, and 81.2 keV) with a total dose of 2.63×10^{15} ions/cm², corresponding to a peak Er concentration of 3.25×10^{20} ions/cm³ ($\approx 0.34\%$ atomic ratio in Si₃N₄).

Figure S24(a) shows the measured refractive indices $n(\lambda)$ before and after implantation. Implanting 0.34% Er increases the film index slightly, from 1.9866 to 2.0106 at 1552 nm, indicating a small modification of material dispersion $|dn/d\lambda|$. To assess the optical impact on waveguides, we measured the integrated dispersion $D_{\text{int}}/2\pi$ of a 22.5 GHz Si₃N₄ ring resonator before and after implantation (Fig. S24(b)). The free spectral range ($D_1/2\pi$) increased from 22.544 GHz to 22.684 GHz, corresponding to a $\sim 0.6\%$ reduction in the group index $n_g = c/(L \cdot \text{FSR})$. This behavior is consistent with the measured index change: although n_{eff} slightly increases, the slope $dn_{\text{eff}}/d\lambda$ decreases, resulting in a lower n_g . Minor geometry or stress variations from implantation and post-annealing may contribute to this small dispersion shift, but the overall effect remains weak. At higher Er concentrations, a gradual increase in the Si₃N₄ refractive index is expected. Further investigation of the concentration dependence will be reported in future work.”

- I believe in Fig 1B Er:Al₂O₃ should be SiN/Er:Al₂O₃

We thank the reviewer for the clarification. We agree that the platform combines Si₃N₄ passive waveguide with Er:Al₂O₃ gain. Accordingly, Fig. 1B (and legend) has been updated from ‘Er:Al₂O₃’ to ‘Si₃N₄/Er: Al₂O₃’.

Action taken:

We change the label in Figure 1B and the corresponding caption from “Er:Al₂O₃” to “Si₃N₄/Er:Al₂O₃”.

- Figure 1 caption: part G - "The effective optical mode areas...are shown for comparison." Aren't these the ion implant energies?

We thank the reviewer for the comment. The markers denote the implantation energies corresponding to the waveguide center for 700 nm [19] and 200 nm (this work) Si₃N₄ waveguides; the caption has been revised for

clarity.

Action taken:

We change the caption in Figure 1G:

“Simulated effective optical mode areas of the fundamental TE mode and required implantation beam energy to reach the waveguide center, as a function of the Si₃N₄ waveguide thickness. For comparison, the implantation energies for 700 nm [19] and 200 nm-thick waveguides (this work) are indicated.”

- an exact absorption range of 1500-1538 nm is given related to hydrogen...this absorption can extend beyond those wavelengths, perhaps modify this slightly or give a reference that explains why this exact range for your material

We thank the reviewer for the clarification. The 1500-1538 nm range refers to the characteristic N-H and Si-H vibrational overtone absorption bands in LPCVD Si₃N₄ films, typically centered near 1501 nm, 1517 nm, and 1530 nm [R12]. High-temperature annealing (~1200 °C) effectively removes hydrogen-related bonds [R13], thereby suppressing absorption within this range (Fig. R8). We have revised the text to clarify that this wavelength range corresponds specifically to N-H/Si-H absorption bands and cited the appropriate reference.

[R12] M. Pfeiffer, et al. Ultra-smooth silicon nitride waveguides based on the Damascene reflow process: fabrication and loss origins. *Optica* 5.7: 884-892 (2018).

[R13] X. Ji, et al. Efficient mass manufacturing of high-density, ultra-low-loss Si₃N₄ photonic integrated circuits. *Optica* 11.10: 1397-1407 (2024).

Action taken:

- We have replaced the “1500-1538 nm” range in the revised main text:

“Post-deposition annealing at 1200 °C eliminates hydrogen-related defects [33], minimizing absorption in the 1500-1530 nm range [34][35].”

[33] W. Jiang, et al. Effect of hyperthermal annealing on LPCVD silicon nitride. *Materials Science in Semiconductor Processing* 43: 222-229 (2016).

[34] M. Pfeiffer, et al. Ultra-smooth silicon nitride waveguides based on the Damascene reflow process: fabrication and loss origins. *Optica* 5.7: 884-892 (2018).

[35] X. Ji, et al. Efficient mass manufacturing of high-density, ultra-low-loss Si₃N₄ photonic integrated circuits. *Optica* 11.10: 1397-1407 (2024).

- I suggest giving the FSRs and Vernier FSR in terms of wavelength in brackets as well for easy reference

We appreciate the reviewer’s suggestion to clarify the Vernier FSR. We now report the FSRs and Vernier FSR in wavelength as well as frequency. Using $\Delta\lambda = \frac{\Delta\nu \cdot \lambda^2}{c}$, the 144 GHz and 142 GHz FSRs of resonators correspond wavelength spacings of ≈ 1.15 nm and ≈ 1.14 nm at 1550 nm. The Vernier FSR of 10.224 THz corresponds to ≈ 81.93 nm. The main text and Fig. 2C have been updated accordingly (Fig. 2C already shows both frequency and wavelength axes).

Action taken:

We have added the FSRs in wavelength domain in the main text:

“The microresonators designed with FSRs of 144 GHz (≈ 1.15 nm) and 142 GHz (≈ 1.14 nm) result in a Vernier FSR of 10.224 THz (≈ 81.9 nm).”

- Indicate lasing wavelength for laser power measurement in Fig. 3E

We appreciate the reviewer’s comment on the wavelength indication. The lasing slope efficiency measurement was performed at 1591.9 nm. The caption of Fig.3E has been updated accordingly.

Action taken:

- We have added the operation wavelength in Fig. 3E caption:

“Dependence of lasing power on pump power, both measured on-chip, showing a 17.5 mW threshold and ~24% slope efficiency at 1591.9 nm (Device ID: D125_04_F4_C3).”

- In main text:

“Figure 3E presents the on-chip laser power as a function of on-chip pump power, revealing a lasing threshold of 17.5 mW at 1480 nm pumping and a slope efficiency of approximately 24% at 1591.9 nm. Supplementary Note 6 presents simulations and measurements of slope efficiency as a function of wavelength, showing higher efficiency across 1560–1600 nm rather than at the Er³⁺ emission peak, due to reduced reabsorption and lower passive loss. Further improvements are anticipated by reducing cavity and Vernier-filter insertion losses and mitigating Er-ion quenching.”

- Perhaps it is a preference, but in general it might be preferred to have plot axes begin and end on specific labelled values for readability. Perhaps this is ok for the journal as is, but Fig. 4A and C, for example, particularly stand out as looking like they are missing an x axis label on the left

We thank the reviewer for this readability suggestion and agree. In the original rendering, the leftmost X-axis tick label in Figs. 4A and 4C was suppressed during export to avoid overlap. We have regenerated these panels with explicit axis limits and tick positions so that each axis begins and ends on labeled values.

Action taken:

Figs. 4A and 4C now use rounded bounds with labeled end ticks. No data were changed.

Figure R20: Revised Figure 4 with labeled end ticks.

- in the conclusion, the first Er-doped waveguide laser fabricated by wafer scale processes is claimed - this should be refined a bit since others have demonstrated wafer-scale on-chip Er lasers. E.g. first Er-implanted SiN wafer scale laser. In fact the statement in the abstract should also be refined a bit, since foundry compatible tunable Er waveguide lasers have been shown in the past.

We thank the reviewer for this helpful clarification. We agree that wafer-scale and foundry-compatible on-chip Er-doped lasers have been demonstrated previously, predominantly with Er:Al₂O₃ gain on Si/SiN platforms. Our claim is now refined to specify the novelty of this work: the first wafer-scale, foundry-compatible Er-implanted Si₃N₄ tunable waveguide lasers. We have revised the abstract and conclusion accordingly and added citations to prior wafer-scale/foundry-compatible Er:Al₂O₃ lasers for context.

Action taken:

- We revise the statement of first foundry-compatible wafer scale claim to:

“Here, we overcome these limitations and demonstrate the first fully wafer-scale, foundry-compatible Er-doped Si₃N₄ photonic integrated circuits-based tunable lasers.”

- We have revised the main-text claim to specify broadband C+L operation on an Er-implanted Si₃N₄ platform and to acknowledge prior wafer-scale Er-laser demonstrations:

“Prior wafer-scale rare-earth lasers deposited Er:Al₂O₃ on Si or Si₃N₄ waveguides [20, 21], and mode-hybrid Tm:Al₂O₃/Si₃N₄ large mode area (LMA) amplifiers and lasers have scaled on-chip power [22, 23], but neither is a monolithic, directly implanted Er:Si₃N₄ platform.

Here we directly implant Er into Si₃N₄ waveguides and demonstrate wafer-scale, C+L-band tunable waveguide lasers with fiber-laser-class coherence without external seeding [24].”

[20] Purnawirman, et al. C-and L-band erbium-doped waveguide lasers with wafer-scale silicon nitride cavities. Optics letters 38.11: 1760-1762 (2013).

[21] N. Li, et al. Monolithically integrated erbium-doped tunable laser on a CMOS-compatible silicon photonics platform. Optics Express 26.13: 16200-16211 (2018).

[22] N. Singh, et al. Towards CW modelocked laser on chip—a large mode area and NLI for stretched pulse mode locking. Optics express 28.15, 22562-22579 (2020).

[23] N. Singh, et al. Watt-class silicon photonics-based optical high-power amplifier. Nature Photonics 19.3, 307-314 (2025).

[24] N. Singh, et al. Sub-2W tunable laser based on silicon photonics power amplifier. Light: Science & Applications 14.1,18 (2025).

- the Methods section seems a bit unnecessary, since it is very short and fabrication details also appear in the main body and supplementary material. I suggest condensing this into two places instead of three and eliminating the Methods.

We respectfully disagree. The main text provides a high-level, figure-linked overview, while the Methods compiles all parameters needed for reproducibility. Specifically, Methods reports film thickness and uniformity, anneal temperatures and durations, DUV lithography specifications, dry-etch chemistries, post-Er-implant treatments, ICPCVD cladding precursors and conditions, the microheater material stack, and chip singulation. The Supplementary Information (Sec. S2) serves a different purpose: it analyzes thickness shrinkage and bending in large-aspect-ratio waveguides and analyze its potential impact on device loss and dispersion, and motivates the deposition → anneal → etch sequence (as opposed to the reversed order used in many Si₃N₄ lines). Condensing Methods into the main text would lengthen and fragment the narrative, while removing it would compromise reproducibility. Nevertheless, we agree to streamline the manuscript by removing minor redundancies.

Action taken:

We trimmed minor redundancies between the main text and Methods to keep the narrative concise while preserving a reproducible process description.

- Revised sentence: “Following deposition, Si₃N₄ films are annealed at 1200 °C for 11 hours to eliminate excess H₂ and break N-H and Si-H bonds, ~~which exhibit absorption in the 1500–1538 nm wavelength range.~~”
- Removed the duplicate description of the post-Er-implantation treatment.

- Supplementary material: The supplementary material is very helpful and thorough. I just have a few suggestions:

We thank the reviewer for the positive assessment of our Supplementary Information.

-> Fig S5: indicate that it is the cladding or SiO₂ in the legend, otherwise it might be mistaken for H in the SiN (which is a typical consideration)

We thank the reviewer for this suggestion. We now label the material explicitly. The legend and caption now state that the absorption is in the SiO₂ cladding.

Action taken:

- We updated Fig. S5 legend to: “P_{out} w/o H absorption in SiO₂”, “P_{out} w/ H absorption in SiO₂”, and “P_{out} with 1 dB/m constant loss”.
- We clarified in the caption: “This simulation also considers SiO₂ cladding with H-related absorption near 200 THz”.
- We refined the simulation using a rate-equation + propagation model (detailed in the revised Supplementary Section 4) and replaced Fig. S5 with the updated results:

Figure R9: **Parametric sensitivity of the output power at 1550 nm.** (a) Output power vs passive loss α (dB/m) with Vernier filter transmission $T_f = 0.9$, $R_1 = 1$, $R_2 = 0.2$, non-emitting (parasitic) fraction $n_c = 0.1$. The pump loss is fixed to 1 dB/m. (b) Output power vs Vernier filter transmission T_f , $\alpha = 1$ dB/m, $R_1 = 1$, $R_2 = 0.2$, $n_c = 0.1$. (c) Output power vs R_1 , with $\alpha = 1$ dB/m, $T_f = 0.9$, $R_2 = 0.2$, $n_c = 0.1$. (d) Output power vs R_2 , with $\alpha = 1$ dB/m, $T_f = 0.9$, $R_1 = 1$, $n_c = 0.1$. (e) Output power vs non-emitting ion

fraction n_c , with $\alpha = 1$ dB/m, $T_f = 0.9$, $R_1 = 1, R_2 = 0.2$. (f) Output power vs Er concentration N_0 , with $\alpha = 1$ dB/m, $T_f = 0.9$, $R_1 = 1, R_2 = 0.2$, for $n_c = 0, 0.05$ and 0.1 . (a–f) assume the implanted Er concentration $N_0 = 5.6 \times 10^{26} \text{ m}^{-3}$. (g) Simulated wavelength-dependent output power, based on measured passive loss and Vernier transmission from fabricated devices (blue trace, loop mirrors: $R_1 = 1, R_2 = 0.2, n_c = 0.1, N_0 = 5.6 \times 10^{26} \text{ m}^{-3}$), comparing with a constant propagation loss of 1 dB/m (red trace). This simulation also considers SiO_2 cladding with H-related absorption near 200 THz (yellow trace, with pump loss of 20 dB/m), and an ideal case (green trace, $\alpha = 1$ dB/m, $T_f = 1$, $R_1 = 1, R_2 = 0.2, n_c = 0, N_0 = 5.6 \times 10^{26} \text{ m}^{-3}$). (h) Output power as a function of gain spiral length, based on measured passive loss and Vernier transmission from fabricated devices, showing an optimum around 0.08–0.1 m at an Er concentration of $N_0 = 5.6 \times 10^{26} \text{ m}^{-3}$, with n_c varying from 0 to 0.1.

-> Section 4: perhaps I'm missing it, but in case not, give the cavity roundtrip length value

The cavity round-trip length is $L \approx 40.14$ cm (indicated in the intracavity power-evolution simulation in Fig. S5(b)). For completeness, we also quantify the cavity delay to relate the physical length to the measured spectral spacing. Using $n_g = 1.8$ at 1550 nm, the calculated round-trip time is $\tau = \frac{n_g L}{c} \approx 2.41$ ns. However, the measured longitudinal mode spacing ($\Delta\nu = 300$ MHz) yields $\tau = \frac{1}{\Delta\nu} \approx 3.33$ ns, i.e., 0.92 ns longer due to group delay from the Vernier microrings (as noted in Sec. 4).

In the updated Supplementary Sec. 7 (TRN transduction), we derived the phase added by the Vernier rings:

$$\varphi_{\text{vernier}} = -\arctan\left(\frac{\Delta_1}{\kappa_1}\right) - \arctan\left(\frac{\Delta_2}{\kappa_2}\right)$$

where $\Delta_j = \omega_j - \omega$ are the detunings from each ring resonance and $\kappa_j = \kappa_{0,j}/2 + \kappa_{\text{ex},j} + \kappa_{\text{p},j}$ denote the total decay rates. The additional group delay during each cavity round trip is:

$$\tau_{\text{vernier}} = 2 \frac{d\varphi_{\text{vernier}}}{d\omega} = \frac{2/\kappa_1}{1 + (\Delta_1/\kappa_1)^2} + \frac{2/\kappa_2}{1 + (\Delta_2/\kappa_2)^2}$$

Using $\kappa_1 = \kappa_2 \approx 0.6 \text{ GHz} \cdot 2\pi$, the calculated round-trip delay map versus (Δ_1, Δ_2) is shown in Fig. R21:

Figure R21: Additional single-pass delay from the Vernier filter as a function of detuning.

At dual resonance ($\Delta_1 = \Delta_2 = 0$), the Vernier filter adds a delay of ~ 1 ns to the cavity round-trip time, consistent with the 0.92 ns inferred from the measured mode spacing.

Action taken:

- We added the explicit values to the updated Supplementary Section 6:

“... L is the cavity round-trip length (~ 40.14 cm), g is the gain coefficient ($1/m$), and α is the propagation loss ($1/m$).”

- We added the Vernier filter group-delay derivation in a new Supplementary Section “**Group delay of Vernier resonators**”:

“In this section, we quantify the group delay introduced by the Vernier microrings to account for the discrepancy between the geometric round-trip time and that inferred from the measured mode spacing ...”

-> elaborate on what is meant by "wavelength-independent passive loss"

By *wavelength-independent passive loss* we mean that the propagation loss is assumed constant across the band, i.e. $\alpha(\lambda) = \alpha_0$, rather than increasing toward shorter wavelengths due to Rayleigh scattering and absorption.

Action taken:

We state explicitly $\alpha(\lambda) = 1$ dB/m.

-> the following is somewhat confusing: "The L-band EDWL output power matches the simulated value, indicating a smaller O-H absorption tail in real devices than initially modeled."

We thank the reviewer for noting the ambiguity. Our intent is that the simulated L-band output without O-H absorption matches the measurement, indicating that any O-H absorption tail into the L band is weaker than initially assumed. We have updated the output power simulation in Supplementary Section 6 (Fig. R9).

Action taken:

For clarity, we removed the sentence and state the result directly.

-> I strongly suggest including an experimental diagram/diagrams illustrating the setup and how the measurement is done for section 5 (e.g. in Figure S7) since it is a bit hard to follow

We agree that a schematic will improve readability. We briefly describe the measurement of pump-RIN-to-EDWL frequency-noise transduction: We inject a small sinusoidal current $i(t) = I_{\text{bias}} + \Delta I \sin(2\pi f_m t)$ into the 1480-nm DFB pump using the AWG. We first measure the resulting pump RIN tone at f_m with a photodetector, the RIN is recorded with an ESA, yielding a calibrated fractional modulation $m = \Delta I / I_{\text{bias}}$. The modulated pump then drives the EDWL. Via thermo-optic effect this intensity modulation is converted to laser frequency fluctuations. We measure the EDWL frequency-noise spectrum $S_{\delta\nu}(f)$ using delayed self-heterodyne interferometry. Narrow peaks at f_m in $S_{\delta\nu}(f)$ quantify the RIN to FN transduction; sweeping f_m (2–400 kHz) maps the frequency dependence to obtain the transfer function $S_{\delta\nu}(f) = H(f) \cdot \text{RIN}(f)$. We keep $m \ll 1$ (here 5.7×10^{-3}) to ensure linear, small-signal conditions.

Action taken:

We added an experimental schematic in revised Fig. S7 illustrating the measurement setup for pump-RIN-to-EDWL frequency-noise transduction.

We have revised the text to clarify the experiment:

“We quantified the transduction from pump-laser intensity modulation to the EDWL frequency noise using the setup in Figure S14(a). An arbitrary waveform generator (AWG) applied a small sinusoidal current to the 1480-nm pump diode driver:

$$i(t) = I_{\text{bias}} + \Delta I \sin(2\pi f_m t)$$

Figure R22: Revised Figure S7. (a) Experimental setup for the pump RIN to the EDWL frequency noise transduction. An AWG applies a sinusoidal current $i(t) = I_{\text{bias}} + \Delta I \sin(2\pi f_m t)$. The two paths: (I) measures the pump RIN with a photodiode and an electrical spectrum analyzer. The EDWL output (II) is analyzed by delayed self-heterodyne interferometry. (b) Pump RIN measurement for modulation index $m = \frac{\Delta I}{I_{\text{bias}}} = 5.7 \times 10^{-3}$ at $f_m = 2, 10, 50, 100, 150,$ and 400 kHz. An AC-coupled pre-amplifier is inserted between the PD and ESA. (c) The corresponding EDWL frequency noise spectral density from DSHI. Discrete peaks show the RIN to FN transduction.

with $I_{\text{bias}} = 1400$ mA. The driver's RF input has a modulation transconductance $C_{\text{mod}} = 200$ mA/V and a bandwidth of 1.2 MHz. For an AWG setting $V_{\text{pp}} = 40$ mV, the current amplitude $\Delta I = \frac{1}{2} C_{\text{mod}} V_{\text{pp}} = 4$ mA. The calibrated fractional modulation is $m_{\text{pp}} = 2\Delta I / I_{\text{bias}} \approx 5.7 \times 10^{-3}$. With low modulation depth and within the modulation bandwidth, the relationship between current modulation and laser frequency noise was assumed to be linear. This was verified by the absence of harmonic peaks on the RIN spectrum after applying the modulation.

At a few modulation frequencies f_m from 2 kHz to 400 kHz, we measured both the pump RIN and laser frequency noise (Figure S14(b)(c))....”

-> section 6 can use a few more sentences explanation explaining defining thermal refractive noise, perhaps a reference such as the authors' own work and how it manifests in resonators and how temperature plays a role at the beginning, since temperature doesn't appear in the math

We thank the reviewer for this suggestion. Thermorefractive noise (TRN) is the fundamental refractive index fluctuation in a dielectric caused by thermodynamic temperature fluctuations of the material. In a microresonator, random, equilibrium fluctuations δT within the optical mode change the refractive index via the thermo-optic coefficient dn/dT , thereby jittering the cavity resonance frequency f_0 . At thermal equilibrium, the variance of temperature fluctuations averaged over a volume V is set by [R14]:

$$\langle \delta T^2 \rangle = \frac{k_B T^2}{\rho C V}$$

where T is the absolute temperature, k_B the Boltzmann constant, ρ the material density and C the specific heat. The fractional frequency variation is $\frac{\sqrt{\langle \delta f^2 \rangle}}{f} = \frac{\sqrt{\langle \delta T^2 \rangle}}{n} \frac{dn}{dT}$, which is proportional to T .

In both models of effective temperature fluctuations in the microresonator in [R14], $S_{\delta T}(\omega) \propto T^2$. The frequency noise spectral density is related to temperature fluctuations by $S_{\delta f} = (f_0 \frac{df}{dn} \frac{dn}{dT})^2 S_{\delta T} \propto T^2$.

Accordingly, Section 6 specifies a conservative assumption by evaluating TRN with both Vernier rings at 400 K. This slightly overestimates the transduced frequency noise because, in the experiment, one ring is held near room temperature. Since the TRN-induced frequency noise PSD scales as $S_{\delta f} \propto T^2$, the untuned ring's contribution at 298 K is lower than at 400 K.

[R14] G. Huang, et al. Thermorefractive noise in silicon-nitride microresonators. *Physical Review A* 99.6: 061801 (2019).

Action taken:

We added the description of thermorefractive noise and a reference in Supplementary Section X:

“Thermorefractive noise (TRN) is the fundamental refractive index fluctuation in a dielectric caused by thermodynamic temperature fluctuations of the material. In a microresonator, random, equilibrium fluctuations δT within the optical mode change the refractive index via the thermo-optic coefficient dn/dT , thereby jittering the cavity resonance frequency ω_0 ...

... where $S_{\delta f_1}$ is the frequency noise power spectrum density of a single Vernier ring due to TRN. In microresonators, the frequency noise PSD relates to temperature fluctuations by $S_{\delta f} = (f_0 \frac{df}{dn} \frac{dn}{dT})^2 S_{\delta T} \propto T^2$ [S14]. In the approximation of Eq (19), we assume the two Vernier rings to have the same temperature at 400 K, although this slightly overestimates the transduced frequency noise, as in experiments one of the rings is maintained at room temperature.”

*[S14] G. Huang, et al. Thermorefractive noise in silicon-nitride microresonators. *Physical Review A* 99.6: 061801 (2019).*

-> Section 7: add setup schematic for heterodyne interferometry or refer to main text if it is there

We thank the reviewer for the suggestion. The heterodyne interferometry setup is shown in the main text (Fig. 3A), where the EDWL output is combined with a narrow-linewidth CW reference, and the beat note is detected on a high-speed photodiode and analyzed with an ESA to extract the frequency noise spectrum.

Action taken:

The heterodyne measurement setup is documented in Fig. 3A and now serves as the reference for the method described in the Supplementary Information.

“Figure S16(j) reports the EDWL frequency-noise spectra (heterodyne measurement with a low-noise Topica ECDL, setup in Fig. 3A in the main text) ...”

- it would be good to check for grammar/typos, I spotted the following:

-> a 8 μm -> an 8 μm

-> on ultra-low loss Si3N4 PICs fabrication -> on the ultra-low loss Si3N4 PICs fabrication

-> supplementary material: norminal -> nominal

-> supplementary material: in EDWL -> in the EDWL or in EDWLs

We thank the reviewer for the careful revision of our grammar errors. The spotted errors are corrected in the revised manuscript.

Action taken:

We corrected the spotted grammar errors:

- a 8 μm \rightarrow an 8 μm
- on the ultra-low loss Si_3N_4 PICs fabrication
- norminal \rightarrow nominal
- We avoided writing “EDWL” with no article.

Reviewer #3

The authors have demonstrated an integrated design and operation of Er-ion doped Si-nitride based tuneable laser device operating in the C+L band. The temperature tuneability and noise figures see quite impressive. However, there are several comments which might help the authors to improve the quality of paper and also set the scene for future development.

We appreciate the reviewer's positive assessment and constructive suggestions for improving the manuscript.

a) The main improvement in the Si₃N₄-based waveguide is the quality of deposited films and low-OH loss in silica. When silica mixes with Si₃N₄, it forms silicon oxynitride and the OH-ions therein at the interface will overlap with Er-ion absorption band, and thereby contribute to OH-concentration quenching. There is no data to show how much reduction in OH was achieved as a result of the process control. Was it the high 2.0 MeV the cause of larger OH-absorption or, something else. This is unclear.

Figure R23: (a) Near-infra-red absorption spectrum of water [R15]. (b) Broadband linewidth measurement of Si₃N₄ resonators with O-H related absorption.

We thank the reviewer for the insightful comment. O-H absorption in SiO₂ typically peaks near 1.45 μm with a tail extending into the C band (Fig. R23(a) and [R15]), and can in principle reduce gain or contribute to quenching. This absorption originates from hydrogen incorporated in the oxide network or moisture in low-temperature, porous SiO₂ films, not from ion implantation: implantation (energies <500 keV) is applied to uncladded Si₃N₄ and followed by SiO₂ deposition, so ion implantation is not a source of O-H absorption. In addition, our Si₃N₄ waveguides are fabricated by LPCVD and annealed at 1200 $^{\circ}\text{C}$ for 11 h to remove Si-H and N-H bonds, while the SiO₂ cladding is deposited by ICPCVD at 300 $^{\circ}\text{C}$ using hydrogen-free SiCl₄/O₂ precursors, yielding a broad low-loss window (1260-1625 nm) [R16]. Consistent with this process, Fig. 2D confirms ultra-low loss with no sign of hydrogen-induced absorption.

Concerning silicon oxynitride formation at the Si₃N₄/SiO₂ interface: interdiffusion/oxidation that yields SiO_xN_y requires elevated temperatures ($\geq 800\text{-}1000$ $^{\circ}\text{C}$) [R17]. At 300 $^{\circ}\text{C}$ deposition temperature, such reactions are negligible. Moreover, Er ions are confined to the Si₃N₄ core to maximize modal overlap (Fig. 1F). Any O-H in SiO₂, or hypothetical SiO_xN_y, if present, resides at or outside the Si₃N₄ boundary and does not spatially overlap the Er distribution; thus, O-H-induced concentration quenching of Er is not expected.

To verify process quality, ATR spectra (Fig. R24; [R16]) show that SiCl₄/O₂-deposited SiO₂ lacks the characteristic O-H bands seen in SiH₄-based oxides, confirming a low-OH, stable, and repeatable process.

In the manuscript we attributed the observed C-band power penalty to O-H absorption in the SiO₂ cladding and identify residual hydrogen contamination (e.g., cross-contamination in the ICPCVD chamber) as the cause. We

Figure R24: Near and mid infrared optical absorption of SiO_2 deposited with different processes. Source: [R15].

further reported this O-H absorption in Supplementary Section 11, attributing it to water exposure; a pronounced loss peak near 1420 nm in a test resonator (Fig. R23(b)) is consistent with water-related absorption. However, subsequent measurements revealed that this peak is not ubiquitous across the wafer: all test resonators share the same $5 \mu\text{m} \times 200 \text{ nm}$ cross-section, but only devices with sub-micron bus-ring gaps exhibit the O-H peak; it is absent in devices with gaps larger than $1 \mu\text{m}$ (e.g., the resonator in Fig. 2D was measured on the same wafer as Fig. R23(b)). We therefore attribute the peak to residual water trapped in small features; this mechanism is irrelevant to our EDWL, which has no sub-micron gaps. The reduced C-band lasing power is instead explained by device-level effects that we now clarify: the Vernier filter FSR is 10 THz, which is smaller than the $\sim 12 \text{ THz}$ Er emission bandwidth, so resonances around $\sim 1530 \text{ nm}$ and $\sim 1610 \text{ nm}$ can overlap; combined with a higher net gain (lower propagation loss and smaller reabsorption), the L-band preferentially wins the gain competition. Figure 3D in the manuscript shows near-full C+L lasing when the WDM loop mirror is biased to suppress L-band lasing; however, this bias also increases C-band cavity loss. We have revised the text to reflect this analysis and to note the water-trapping effect in Supplementary Section 11.

[R15] T. Afrin, et al. Water structure modification by sugars and its consequence on micellization behavior of cetyltrimethylammonium bromide in aqueous solution. *Journal of Solution Chemistry* 42.7: 1488-1499 (2013).

[R16] Z. Qiu, et al. Hydrogen-free low-temperature silica for next generation integrated photonics. *arXiv preprint arXiv:2312.07203* (2023).

[R17] S. Taguchi, et al. Silicon nitride oxidation behaviour at 1000 and 1200 °C. *Journal of materials processing technology* 147.3: 336-342 (2004).

Action taken:

- We have revised the main text to explicitly state that the SiO_2 cladding is deposited by hydrogen-free SiCl_4/O_2 ICPCVD at 300 °C, which minimizes O-H absorption and precludes SiO_xN_y formation at this temperature:

“A $3 \mu\text{m}$ hydrogen-free, low-loss SiO_2 cladding is deposited using SiCl_4 and O_2 precursors at 300 °C, thereby minimizing O-H absorption [37] and precluding SiO_xN_y formation [38].”

[37] Z. Qiu, et al. Hydrogen-free low-temperature silica for next generation integrated photonics. *arXiv preprint arXiv:2312.07203* (2023).

[38] S. Taguchi, et al. Silicon nitride oxidation behaviour at 1000 and 1200 °C. *Journal of materials processing technology* 147.3: 336-342 (2004).

- We clarify the origin of the reduced C-band lasing power:

“Nevertheless, a decrease in C-band power relative to the L-band is observed despite stronger Er emission near 1530 nm. This is primarily due to the limited Vernier FSR ($\approx 80 \text{ nm}$; Fig. 2C), which co-aligns C- and L-band resonances (e.g., 1530/1610 nm) and biases lasing toward the L-band where passive loss and absorption

are lower, yielding higher net gain $g - \alpha$. The near-full C+L lasing in Fig. 3D is obtained by bidirectional pumping to raise inversion and by biasing the WDM loop mirror to suppress L-band modes; however, this also increases the loss seen by co-aligned C-band modes, limiting the output power.”

- We also note potential O-H absorption in sub-micron features due to residual water and include Fig. R23 in the Supplementary Information:

“Hydrogen-related bonds (N-H, Si-H) intrinsic to LPCVD Si_3N_4 increase loss in the 1500-1530 nm band [S21] but can be largely eliminated by long-duration high-temperature annealing [S1]. When Si_3N_4 is cladded with SiO_2 , residual water from wet cleaning can be trapped in confined features (e.g., submicron bus-ring gaps) and, if not adequately removed, gives rise to additional O-H absorption in the same spectral range (Fig. S18(a), [S20]) and suppress C-band output in our EDWL. This O-H-related absorption was observed in test resonators with small features. In Fig. S18(b), we quantify the wavelength-dependent passive loss via broadband intrinsic linewidth measurements of a Si_3N_4 microring ($5 \mu\text{m} \times 200 \text{ nm}$) with SiO_2 cladding and a $1 \mu\text{m}$ bus-ring gap. The intrinsic linewidth $\kappa_0(\lambda)$ extracted from resonance fits was converted to propagation loss (dB/m) using $\alpha = 10 \log_{10}(e) n_g \kappa_0 / c$, yielding a loss at 1500 nm approximately twice that at 1600 nm. This confirms an additional O-H contribution, likely from residual water in the small coupling gap. The excess O-H-related loss is concentrated over $\sim 1500\text{-}1550 \text{ nm}$ with only a weak long-wavelength tail. The effect occurs only in devices with bus-ring gaps smaller than $1 \mu\text{m}$; devices of identical cross-section but larger gaps show no added loss (Fig. 2D). The EDWL employs gaps larger than $1 \mu\text{m}$ and should therefore remain unaffected by this absorption.”

[S21] M. Pfeiffer, et al. Ultra-smooth silicon nitride waveguides based on the Damascene reflow process: fabrication and loss origins. *Optica* 5.7: 884-892 (2018).

[S1] X. Ji, et al. Efficient mass manufacturing of high-density, ultra-low-loss Si_3N_4 photonic integrated circuits. *Optica* 11.10: 1397-1407 (2024).

[S20] T. Afrin, et al. Water structure modification by sugars and its consequence on micellization behavior of cetyltrimethylammonium bromide in aqueous solution. *Journal of Solution Chemistry* 42.7: 1488-1499 (2013).

b) Did the structure of Si_3N_4 change when the implantation energy was dropped from 2.0 MeV to 0.5 MeV. The higher implantation energy will also allow Si_3N_4 to implant deeper in the silica layer and contribute to OH-induced loss.

Lowering the implantation energy from 2.0 MeV to 0.5 MeV reduces both the projected range and the damage density in Si_3N_4 , so any structural modification is correspondingly smaller. The cross-sectional SEM image in Fig. 1E shows implanted waveguides without voids or deformation, in contrast to the distortions reported for 2 MeV implants [R3].

SRIM simulation in Fig. 1F shows that the implanted Er ions are fully stopped in the Si_3N_4 layer and do not penetrate the bottom SiO_2 cladding. A narrow lateral strip of SiO_2 adjacent to the Si_3N_4 waveguides is indeed exposed during implantation, so a small Er dose enters that region. This, however, has no impact: the waveguide mode is confined to the Si_3N_4 core and its lateral evanescent field in that SiO_2 strip is negligible (Fig. 1E), yielding a vanishing overlap factor. Any resulting absorption, if present, is far below our loss floor and has no measurable impact on the EDWL. Therefore, the 0.5 MeV implantation does not modify the Si_3N_4 structure nor introduce measurable O-H-related loss.

[R3] Y. Liu, et al. A photonic integrated circuit-based erbium-doped amplifier. *Science* 376.6599: 1309-1313 (2022).

Action taken:

We emphasize the absence of implantation-induced waveguide deformation in the main text:

“It reduces Er implantation time from tens of hours for a $2 \times 2 \text{ cm}^2$ area to tens of minutes for 12-inch wafers (Supplementary Note 1) and minimizes waveguide deformation compared to high-energy methods [18], as verified by cross-sectional SEM image in Figure 1E.”

[18] Y. Liu, et al. A photonic integrated circuit-based erbium-doped amplifier. Science 376.6599: 1309-1313 (2022).

c) The authors demonstrate spontaneous emission spectrum of the $\text{Si}_3\text{N}_4/\text{SiO}_2$ film in Figure 2B is measured with an Argon-ion laser, which is usually at 514nm and not at 520nm? This excitation wavelength also shows the emission features in the visible and near-IR which are important to include.

We thank the reviewer for the clarification. The photoluminescence in Fig. 2B was measured by Prof. Carsten Ronning’s group at Jena University. Upon confirmation, the excitation source was a 520 nm diode laser (not a 514 nm Ar-ion line). The main text and caption have been corrected.

Visible and near-IR Er^{3+} emissions are well documented but fall outside the scope of our C+L-band laser characterization and are not necessary for interpreting the reported results. We currently lack the instruments to measure these wavelengths, and capturing this emission spectrum simultaneously is not feasible. While we don’t include additional spectra here, we recognize their importance and will address them in future studies.

Action taken:

- We correct the laser information in the main text:

*“Figure 2B presents the photoluminescence (PL) spectra of Er-implanted Si_3N_4 and SiO_2 thin films, excited by a 520 nm **diode** laser which populates the $^4I_{13/2}$ state via non-radiative decay from higher energy levels, avoiding in-band stimulated emission.*

- We also corrected the Figure 2B caption:

*“Measured photoluminescence (PL) spectra of Er-doped Si_3N_4 and SiO_2 films, pumped by a 520 nm **diode** laser to excite the $^4I_{13/2}$ state via decay from higher energy levels, free from in-band stimulated emission. Both PL intensities are area-normalized and scaled for clarity.”*

d) In the context of this paper, the authors have used the 1480nm laser which also overlaps with the Er-ion and potentially with the 1st harmonic of the OH-ion. This data is missing from the article which the authors recognise but the quantifiable data are not included (in the article or supplementary section). The loss of gain at shorter wavelength may be due to OH-quenching and Er-ion absorption and re-emission at longer wavelengths (See comment e).

We appreciate the concern. O-H-induced quenching requires co-location with Er^{3+} (near-field coupling of $^4I_{13/2}$ level to O-H vibrations, [R18-R19]). In our platform, Er^{3+} is confined to the Si_3N_4 core, whereas any O-H, when present, resides in the SiO_2 cladding or in moisture trapped in sub- μm gaps. Thus, if present, O-H contributes to propagation loss, not Er site quenching. We agree that the dominant O-H overtone is centered near 1.42 μm and can spectrally overlap with the 1480 nm pump (Fig. R23(a)), introducing pump absorption when present. As noted in response (a), the pronounced $\sim 1.42 \mu\text{m}$ OH feature was observed only in test resonators with sub-micron bus-ring gaps (consistent with trapped moisture) and is absent in the EDWL geometry, in which all gaps exceed 1.5 μm .

We agree that the reduced short-wavelength ($\sim 1530 \text{ nm}$) output is explained by Er reabsorption together with the Vernier-filter FSR constraint, as detailed in response (a). We have revised the manuscript to state these points explicitly and to discuss the possibility of residual O-H absorption.

[R18] A. Monguzzi, et al. Anharmonic overtones quenching in Er^{3+} complexes. *Synthetic Metals* 159.21-22: 2410-2412 (2009).

[R19] S. Shen, et al. Compositional effects and spectroscopy of rare earths (Er^{3+} , Tm^{3+} , and Nd^{3+}) in tellurite glasses. *Comptes rendus. Chimie* 5.12: 921-938 (2002).

Action taken:

We have added a Supplementary section “**Hydrogen-related absorption and Er^{3+} quenching in Si_3N_4 PICs**” to discuss O-H-induced quenching of Er^{3+} and a justification for why this mechanism is not expected in our devices:

“When Si_3N_4 is cladded with SiO_2 , residual water from cleaning in confined features such as sub-micron bus-ring gaps, can introduce additional O-H absorption in the same spectral range (Fig. S18(a), [S20]) and suppress C-band output in our EDWL. This O-H-related absorption was observed in test resonators with small features. The EDWL employs gaps larger than 1 μm and should therefore remain unaffected by this absorption.

It is well established that O-H groups are efficient non-radiative quenchers of Er^{3+} emission near 1.55 μm : the $^4\text{I}_{13/2} \rightarrow ^4\text{I}_{15/2}$ energy gap can be bridged by one O-H overtone or two fundamentals, shortening the Er^{3+} lifetime and lowering quantum efficiency [S22, S23]. However, such quenching requires near-field coupling, i.e., O-H co-located with Er^{3+} within a few nanometers. In our devices, any O-H, when present, resides primarily in the SiO_2 cladding or as trapped water in small gaps; it therefore contributes to propagation loss rather than Er-site quenching.”

[S20] T. Afrin, et al. Water structure modification by sugars and its consequence on micellization behavior of cetyltrimethylammonium bromide in aqueous solution. *Journal of Solution Chemistry* 42.7: 1488-1499 (2013).

[S22] A. Monguzzi, et al. Anharmonic overtones quenching in Er^{3+} complexes. *Synthetic Metals* 159.21-22: 2410-2412 (2009).

[S23] S. Shen, et al. Compositional effects and spectroscopy of rare earths (Er^{3+} , Tm^{3+} , and Nd^{3+}) in tellurite glasses. *Comptes rendus. Chimie* 5.12: 921-938 (2002).

e) The authors have referred Profs. Desurvire and Zervas text book. (E. Desurvire and M. N. Zervas, Erbium-doped fiber) so I presume that that the original reference (Desurvire, Giles, Simpson and Zyskind) in Optics Letters. The articles published by Desurvire et emphasize the importance of Er- population redistribution due to temperature effect and, therefore, the tuneability. My question is with the temperature control the tuneability might be achieved? would this than also require higher energy to maintain and cool the device to achieve broader tuning range. Also, downstream the amplifier might also have a similar issue?

We appreciate the reviewer’s comments on temperature-induced redistribution of Er^{3+} sub-levels and its implications for the tunability of our EDWL. As established by Desurvire and co-authors, temperature redistributes Er^{3+} populations among the Stark sub-levels of the $^4\text{I}_{15/2}$ and $^4\text{I}_{13/2}$ manifolds (via Boltzmann statistics and the McCumber relation), rendering the absorption/emission cross-section ratio weakly temperature dependent, introducing a modest gain-spectrum tilt, and broadening the homogeneous linewidth. The same physics applies to downstream Er-doped amplifiers but does not pose an issue: In our EDWL, wavelength tuning is achieved exclusively by thermally aligning the Vernier microrings. The 17 cm $\text{Er}:\text{Si}_3\text{N}_4$ gain spiral remains near ambient during tuning; consequently, temperature-driven redistribution in the amplifier is negligible and does not provide a deterministic, wavelength-selective tuning mechanism across the C+L band. We therefore do not rely on temperature-based gain shaping, and the reported tuning range requires no power beyond the ring-heater budget and no active cooling.

On the other hand, we have demonstrated lasing at elevated temperatures up to 125 °C (Fig. 4A), confirming thermal robustness of the device. Elevated temperatures can cause Er^{3+} population redistribution and increased homogeneous broadening, which may slightly reshape the gain spectrum, but these effects do not change the

lasing mechanism or the key results reported here. A quantitative map of tunability versus temperature would require stabilized, calibrated fiber-to-chip coupling across temperature cycles and is outside the scope of this work; we will address it in future studies.

Action taken:

We further clarified in the manuscript that tuning is accomplished thermally in the Vernier rings, while the Er-doped spiral is held near room temperature; thus, temperature-induced Er^{3+} population redistribution does not limit or determine the tuning range:

“Pt/Ti microheaters are employed to tune the laser emission wavelength by aligning the peak transmission of the Vernier filter with a cavity longitudinal mode, while the Er:Si₃N₄ gain spiral remains near ambient during tuning. Consequently, temperature-induced Er^{3+} population redistribution [8] is negligible and does not determine the tuning range.”

[8] E. Desurvire and M. N. Zervas, *Erbium-doped fiber amplifiers: principles and applications* (1995).

f) The key question for me is to understand whether there might be spectral hole creation in the ground and $4I_{13/2}$ states which will be dependent on temperature, so a say 1550.15nm signal might be absorbed and become a longer wavelength signal. The Optics Letter of Desurvire et al discusses the transitions.

We thank the reviewer for raising the possibility of spectral hole burning in our EDWL. We agree that ground- and $4I_{13/2}$ state hole creation can occur: when the EDWL lases near 1550 nm, the narrowband intracavity field can weakly burn a hole in the Er^{3+} absorption/gain at that frequency. However, at room temperature the homogeneous linewidth is broad and Stark-level equilibration is ultrafast, so any hole is shallow and rapidly refilled. In steady state this manifests only as a slight, local reduction of gain at the lasing frequency. Because the lasing mode clamps the gain to threshold (see revised Supplementary Section 4), this small reduction is self-compensated and should have a negligible effect on output power. Importantly, in our EDWL, the lasing wavelength is determined by the Vernier filter, which provides a narrow passband and strongly suppresses adjacent modes, as shown in Fig. 3C in the manuscript. Accordingly, such hole burning neither shifts the lasing wavelength nor induces mode hopping, nor does it convert a 1550 nm signal to longer wavelengths (absorption returns energy as spontaneous emission/ASE or non-radiatively). Therefore, hole burning is too weak to define the lasing wavelength or explain the observed L-band preference.

Action taken:

We discuss the possibility of spectral hole burning in the Supplementary Section 4:

“In our EDWL, spectral hole burning can occur in the ground and $4I_{13/2}$ manifolds. However, because the Er-doped gain waveguide operates near room temperature, any hole burned by the intracavity field is shallow and rapidly refilled owing to the broad homogeneous linewidth and ultrafast Stark-level equilibration, manifesting only as minor local gain compression at the lasing line. As the Vernier filter determines the lasing wavelength and strongly suppresses adjacent modes, such weak hole burning neither shifts the lasing wavelength nor induces mode hopping, and it does not limit the tuning range.”

g) There is a mistake in the paper with reference to 4f electronic level shielding? The Er-ion 4f electrons are shielded by 6s and 5d and NOT 5s and 5p? Please read the article and correct.

We respectively disagree. Our manuscript states that Er^{3+} 4f transitions are “shielded by the filled 5s and 5p electron shells,” which is consistent with Desurvire & Zervas [R20]: “The electronic configuration of a trivalent rare earth is $[\text{Xe}] 4f^{N-1}5s^25p^66s^0 \dots$ the inner 4f electrons are shielded from external fields by the outermost 5s, 5p shells.” This configuration contains no 5d electrons ($5d^0$) and has an unoccupied 6s subshell

(6s⁰). In erbium specifically, N = 12, so Er³⁺ has 4f¹¹, i.e., [Xe] 4f¹¹5s²5p⁶ (implying 5d⁰, 6s⁰). These facts underpin why Er³⁺ 4f-4f lines are narrow and weakly host-dependent (see Chapter 4).

[R20] E. Desurvire and M. N. Zervas, Erbium-doped fiber amplifiers: principles and applications (1995).

Submission of a revised version of manuscript NCOMMS-25-17303B

Wafer-scale manufacturing of ultra-broadband, high-power erbium-doped integrated lasers

Dear Editor,
Dear Reviewers,

We thank the reviewers for their careful reading of the revised manuscript, their constructive suggestions, and their recommendation for acceptance in *Nature Communications*.

Reviewer #3

Thank you for answering all the relevant questions raised by Reviewer 3. The queries have been resolved and incorporated in the article and supplementary information. It would be quite nice to include a high-resolution SEM image with element mapping across the Silica/Silicon Nitride interface. The high-resolution elemental map might be able to show the sharpness of interface and potential changes in the refractive index.

We thank the reviewer for the suggestion. We agree that cross-sectional elemental mapping can help visualize interface abruptness. In our platform, however, high-resolution elemental maps across the SiO₂/Si₃N₄ interface are not expected to provide a quantitatively meaningful constraint on interfacial sharpness or refractive-index variation, because the implanted Er concentration is only ~0.34 at.% in Er:Si₃N₄ (mostly at the waveguide center). At this level, EDS/EELS mapping is typically sensitivity-limited and prone to ambiguity from signal-to-noise and quantification artifacts, particularly when the goal is to infer subtle index changes rather than strong elemental contrast.

Importantly, the manuscript already includes a high-resolution cross-sectional SEM (Fig. 1E), showing a well-defined Si₃N₄ core with a sharp surrounding oxide interface and no observable implantation-induced structural modification for the low-energy (<500 keV) wafer-scale process used here. Moreover, SRIM simulations (Fig. 1F) constrain the Er implantation profile to be confined within the 200-nm Si₃N₄ layer, with a projected range well below the Si₃N₄/SiO₂ interface. This conclusion is also validated by our prior RBS depth profiling of Er-implanted Si₃N₄, which shows no penetration into the underlying oxide (Fig. 1E in [1]). Taken together, these results provide internally consistent evidence that the SiO₂/Si₃N₄ interface is abrupt and that Er remains confined to the Si₃N₄ layer.

[1] Y. Liu, et al. A photonic integrated circuit-based erbium-doped amplifier. *Science* 376.6599: 1309-1313 (2022).

The work on integrated photonics with rare-earth is quite interesting in silicon nitride, and the authors here are leading this effort. They have presented already their works in several high impact journal (Y. Liu et. al. A fully hybrid integrated erbium-based laser, Nature Photonics, 18 2024, Y. Liu et. al. A photonic integrated circuit–based erbium-doped amplifier, Science, 376 (2022)). The current work is quite similar to these previous works and the performance level, foot print etc. and the main claim wafer level demonstration is not properly shown - in terms of yield and chip to chip performance over the entire wafer and from wafer to wafer. See my comments below for detail.

- 1) Authors say in the abstract “The reduced implantation energy marks a crucial advance toward scalable production of Er-doped photonic devices. Meanwhile, the increased optical mode area of low-confinement Si₃N₄ waveguides significantly enhances laser performance and output power.”

Could you please explain why it is not considered obvious that a “thin” film silicon nitride will require less energy implantation. Also increased mode area has been explored recently in integrated photonics for high power output which has not been cited in the manuscript it seems – N. Singh. Light Science Appl. 14, 2025, N. Singh et. al. Nat. Photonics, 19, 2025 etc. But the geometry seems different here so there is some novelty but not entirely new in integrated photonics.

- 2) In the introduction authors say the noise figure is low for Er but that is usually true for fiber, however it's not obvious if the noise figure of Er waveguide amplifier is substantially better than semiconductor amplifiers which are additionally electrically pumped.
- 3) What is the variation in performance from chip to chip in the wafer? It would be useful to know, if the claim is about wafer level performance, the variation and yield for data shown in this work, power, tunability, linewidth etc. Such data is quite important for wafer level demonstration.
- 4) And is the implantation over the entire wafer? If so then doesn't it affect the devices which are passives and co integrated, for example the loop mirrors and rings shown in this work? How is their loss affected? Seems like this will increase the loss quite a lot and impact other devices.
- 5) Also the device foot print is quite large (40 mm²) compared to their previous work 6 mm² (Y. Liu et. al. Nature Photonics 2024).
- 6) Also this work has slightly improved tuning range compared to their previous work (Y. Liu et. al. Nature Photonics 2024), but what is not shown is how this improvement is from device to device, chip to chip on the entire wafer, essentially what is the yield? This is a key metric if the main claim is about wafer level. As semiconductor amplifiers are manufactured at industrial scale with high yield which this work compares to.
- 7) What is the measured coupling loss? Simulation value is only mentioned.
- 8) The loss mentioned on page 3 is from 1.5 to 4 dB/m. Is that before ion implantation or after, and what is the loss before and after annealing?
- 9) Are all the measurements in Fig.3 for different devices with only best performing numbers shown? It seems that is the case as device IDs are different for all. Please mentioned that in the manuscript, because then it seems that the high power device with 40 mW on chip power is not the best for wide tunability etc. This seems to suggest the yield and performance is not comparable to wafer level semiconductor devices.
- 10) Is the laser mode hop free over the entire bandwidth? And the laser seems to be better at 1608 nm rather than at the peak of erbium gain, why is that?
- 11) The slope efficiency is 24% and the author suggest reduction in cavity and propagation loss to improve it, which is true but the loss seems already very low, and the figure in supplementary S6a suggest power increase of only 10 mW for loss decrease from 10 dB/m to 1 dB/m for 10 dB gain (which get's worst for lower gain), so it seems it's not possible to increase the power substantially.
- 12) Fig. 4 has interesting data but it seems not clear what the actual linewidth degradation is, please show it in the freq. spectrum (2 d plot – freq. vs power) at different reflection points (at least show at some important points – low reflection, mid-level reflection, high level reflection).
- 13) Also do you expect nonlinear effect with this CW laser power like four wave mixing in the resonator? Which brings to another question what is quality factor of the resonators used?
- 14) Also the concentration of $5.6 \times 10^{26} \text{ m}^{-3}$ seems a bit high, do you expect quenching effects to kick in which rare-earth ions are known for? It seems that this is not a problem as in the supplementary authors simulate with $9.4 \times 10^{26} \text{ m}^{-3}$, please present experimental data if possible on performance with different level concentration.
- 15) In the supplementary eq.4 is solved in time domain or should it be calculated time independently in the steady state (as the laser is in the steady state), and shouldn't the calculation be as a function of length as shown in the supplementary Fig.S5b. Also are the eq.8 – 18 in the supplementary never explored before (please give reference to these equations).